There are amendments to this paper

# The effects of mutational processes and selection on driver mutations across cancer types

Daniel Temko [1,2,3], Ian P.M. Tomlinson [4], Simone Severini[2,5],
Benjamin Schuster-Böckler [6] & Trevor A. Graham [1]

Epidemiological evidence has long associated environmental mutagens with increased cancer risk. However, links between specific mutation-causing processes and the acquisition of individual driver mutations have remained obscure. Here we have used public cancer sequencing data from 11,336 cancers of various types to infer the independent effects of mutation and selection on the set of driver mutations in a cancer type. First, we detect associations between a range of mutational processes, including those linked to smoking, ageing, APOBEC and DNA mismatch repair (MMR) and the presence of key driver mutations across cancer types. Second, we quantify differential selection between well-known alternative driver mutations, including differences in selection between distinct mutant residues in the same gene. These results show that while mutational processes have a large role in determining which driver mutations are present in a cancer, the role of selection frequently dominates.

[1] Evolution and Cancer laboratory, Barts Cancer Institute, Barts and the London School of Medicine and Dentistry, Queen Mary University of London, Charterhouse Sq, London EC1M 6BQ, UK. [2] Department of Computer Science, University College London, London WC1E 6BT, UK. [3] Centre for Mathematics and Physics in the Life Sciences and Experimental Biology (CoMPLEX), University College London, Gower Street, London WC1E 6BT, UK. [4] Institute of Cancer and Genomic Sciences, University of Birmingham, EdgbastonBirmingham B15 2TT, UK. [5] Institute of Natural Sciences, Shanghai Jiao Tong University, Dong Chuan Road, Minhang District, Shanghai 200240, UK. [6] Ludwig Institute for Cancer Research, University of Oxford, Old Road Campus Research Building, Roosevelt Dr, Oxford OX3 7DQ, USA. Correspondence and requests for materials should be addressed to D.T. (email: daniel.temko.13@ucl.ac.uk) or to B.S.-B. (email: benjamin.schuster-boeckler@ludwig.ox.ac.uk) or to T.A.G. (email: t.graham@qmul.ac.uk)

The question of what causes a cancer to have its specific collection of mutations instead of another remains unanswered. The two evolutionary forces of mutation and selection each provide plausible explanations in their own right: the unique mutational exposures of a tissue may provide a 'bias' towards specific mutations, and evolutionary selection for the functional consequences of mutations is similarly highly context specific. In this study, we investigate the contributions of both mutation and selection in shaping the pattern of cancer-associated 'driver' mutations across cancer types.

Environmental mutagens have long been associated with cancer risk[1–3], but links between mutagens and the generation of specific pathological mutations have remained obscure. A landmark study by Alexandrov et al.[4,5] identified distinct "mutational signatures", each the outcome of distinct mutagenic processes, many of which are attributable to environmental mutagens. Each signature consists of the frequency of mutations in 96 "channels" of somatic single-nucleotide substitution variants (SNVs) in the contexts of the two flanking bases. The study described 21 different mutational signatures, each characterised by different proportions of the 96 types. Subsequently > 30 signatures, many with tumour type-specificity, have been reported[6–11].

The likelihood of acquisition of specific cancer-causing mutations[12], hereafter referred to as 'driver mutations', is dependent on the underlying mutational processes, as the probability of a mutation in a particular channel differs between processes. For example, a previous report has highlighted links between APOBEC-induced mutagenesis and specific driver *PIK3CA* mutations across cancer types[13]. Here, we provide a comprehensive statistical assessment of the relationship between relative mutational process activity and driver mutation acquisition across cancer types.

The strength of selection experienced by a mutation is also expected to influence the frequency at which the mutation is detected in the patient population. If two mutations are equally likely to occur, we reason that the more strongly selected mutation will be found more frequently. Traditionally, it has been convenient to classify mutations found in cancer as drivers or passengers[14], but it is likely that the effects of driver mutations actually lie on a continuum, including both 'mini-drivers' and major drivers[15,16]. However, the relative selective advantages of individual driver mutations have not yet been quantified. Here, we present evidence for differential selection between frequently mutated amino acids within a driver gene by controlling for differences in the sequence-specific mutation rate, in cases where the mutational signatures alone cannot fully explain the spectra of mutations in driver genes. We also explore differential selection between sets of related genes that show patterns of mutational exclusivity.

Together, our analysis quantifies the contributions of both mutation and selection in shaping the spectrum of driver mutations across cancer types. Although variable mutation rates influence driver mutation acquisition, large effects of selection are discernible.

## Results

**Links between mutational processes and driver mutations**. We investigated the correlations between mutational process activity and recurrent driver mutations across cancer types. We reasoned that when a mutational process acts, it makes specific driver mutations, caused by a mutation in a specific channel enriched in the mutational signature of the process, more likely. We therefore tested for a difference in the levels of relative mutational process activity between cancers with and without specific driver mutations (Fig. 1a). The use of signature and individual channel activity information was designed to increase the sensitivity and specificity of the approach. Where the activity of a mutational process was significantly higher in cancers with a mutation of interest compared with those without, we considered it

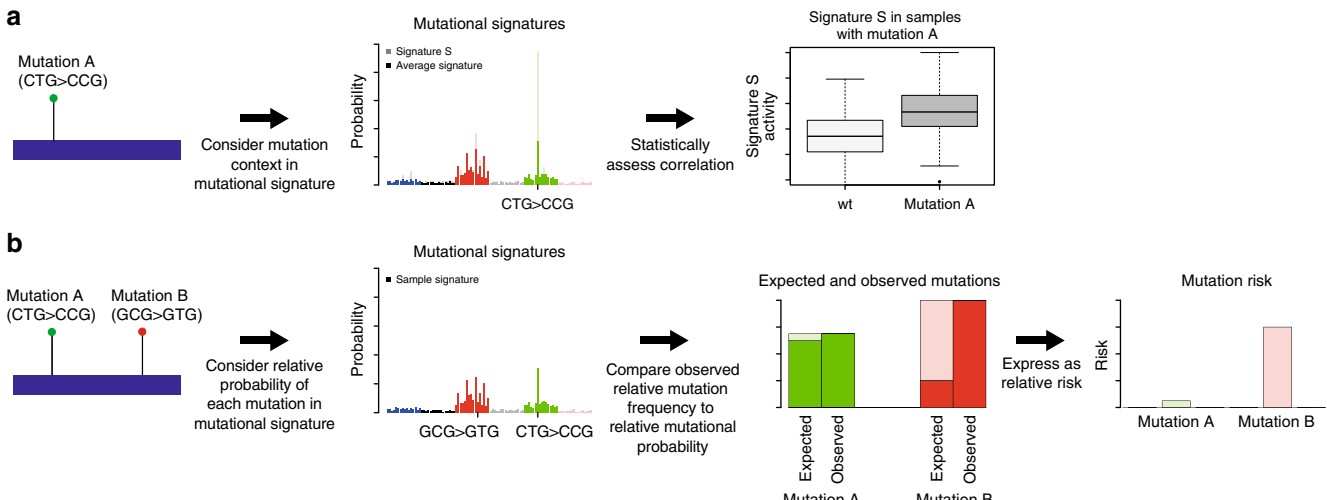

**Fig. 1** Schematic representation of the approach. **a** In the first part of the study, the effects of mutational process activity on driver mutation frequencies were investigated. For a driver mutation, the change was assigned to one of the 96 trinucleotide mutational channels (e.g., CTG > CCG). We hypothesised that mutational signatures in which that channel was higher than average would be over-represented in cancers with these mutations. We tested this hypothesis by comparing the levels of signatures in cancers harbouring the mutations to those in cancers that did not harbour the mutations. **b** In the second part of the study, we investigated the effects of mutational processes on the relative frequencies of specific pathogenic mutations in cancer driver genes. The causal channels of the different driver mutations (different amino-acid changes) within a gene were identified on a tumour type by tumour type basis. We then tested whether observed frequencies of each driver mutation differed significantly from those expected based on mutational process activity alone, thus indicating differential selection between mutations in the same gene. Using a simple mathematical model, we transformed normalised measurements of mutation frequency into estimates of relative risk between mutations. This analysis was then extended to comparisons between mutations in different driver genes with apparently equivalent functional effects in a cancer type

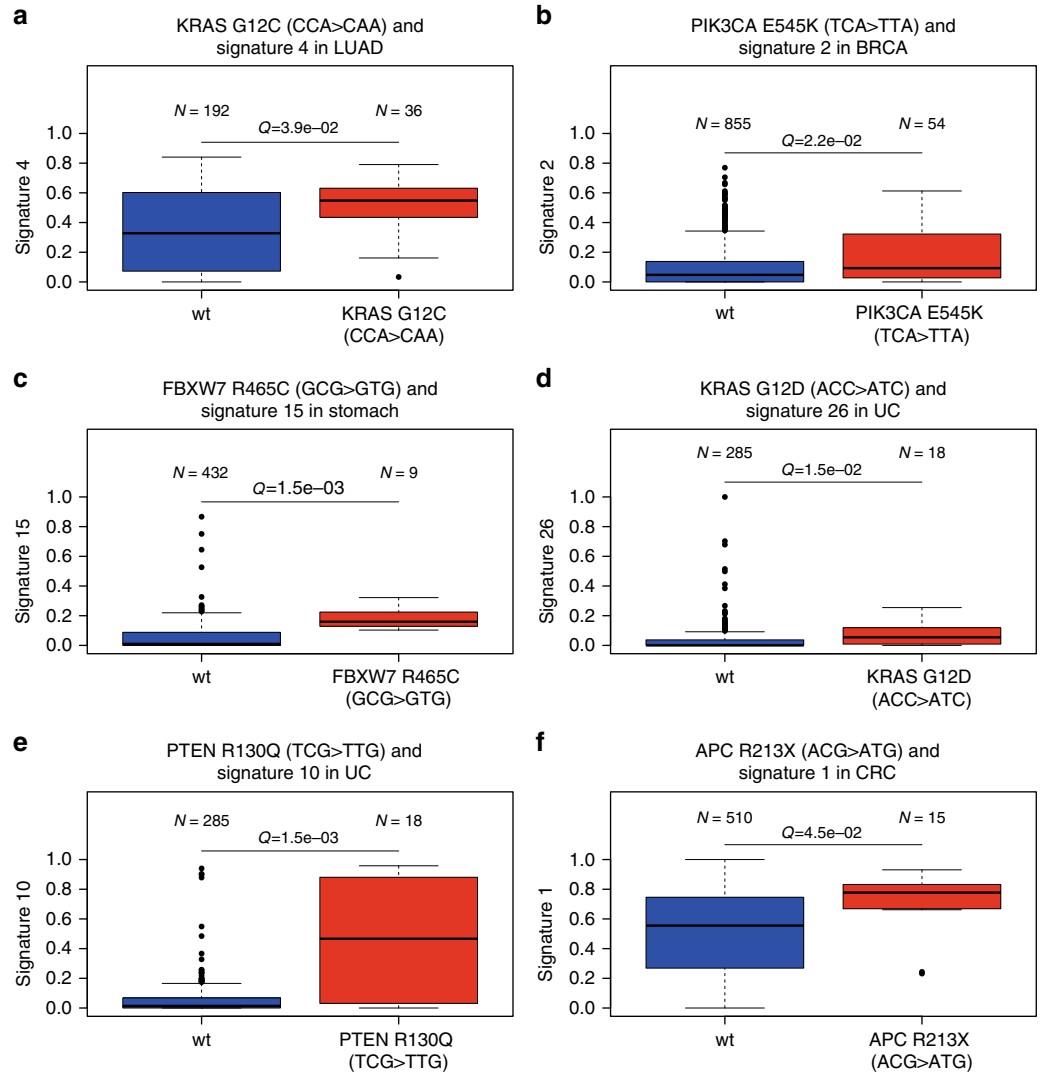

**Fig. 2** Selected associations between mutational process exposures and driver mutations within cancer types. Q-values shown are for Mann–Whitney U-test. **a** *KRAS G12C* and signature 4 in lung adenocarcinoma. **b** *PIK3CA E545K* and signature 2 in breast cancer **c** *FBXW7 R465C* and Signature 15 in stomach cancer **d** *KRAS G12D* and signature 26 in uterine carcinoma **e** *PTEN R130Q* and signature 10 in uterine carcinoma **f** *APC R213X* and signature 1 in colorectal cancer. Centre line shows median; box shows inter-quartile range; upper whisker shows the upper quartile plus one and a half times the inter-quartile range, or the maximum data point (if lower); lower whisker shows the lower quartile minus one a half times the inter-quartile range, or the minimum data point (if higher)

supporting evidence for a causative relationship between the mutational process activity and the acquisition of the driver mutation.

Data were obtained and curated from the TCGA and International Cancer Genome Consortium (ICGC) data portals (see Methods). Driver genes were classified according to a recent study[16]. The data set for analysis represented 11,336 samples across 22 major cancer types (listed in Supplementary Table 1). There were 1447 whole-genome samples and 9889 whole-exome samples. Analysis using only exonic mutations from the whole-genome samples revealed similar relationships between mutational processes and driver mutations (see Methods). Downstream analysis was based on 14,356,672 SNVs, of which 40,753 were non-synonymous mutations in driver genes. We did not consider other types of genome alteration (such as copy number alteration).

We estimated the relative activity (exposure) of each mutational process in each of the samples, using non-negative least squares regression (see Methods). Simulated data showed

that tumours with 20 mutations or more provided sufficiently accurate recovery of mutational processes (see Methods; Supplementary Figs. 1–3). Consequently, we excluded 1153 samples with fewer than 20 mutations from this analysis, leaving 10,183 samples for further analysis. To test for potential signature mis-assignment, we also considered a more stringent cutoff of 50 mutations, which gave similar results (see Methods). In each cancer type, we classed as 'recurrent' non-silent DNA mutations in driver genes as those that occurred at least four times in the cancer type (these recurrent mutations were considered candidate tissue-specific driver mutations). For each mutation, we selected the channel among the 96 possibilities that matched the observed mutation (hereinafter, the 'causal channel' of the candidate driver mutation). For this channel, we identified the signatures where the frequency of the causal channel was above average, relative to all mutational processes active in the cancer type. For each of these signatures, we tested for a correlation between mutational process activity and presence of the mutation in the cancer type.

We analysed the power to detect an association for each of the tests. To do this, we simulated the driver mutation profile for the samples in the cancer type, based on their inferred mutational process exposures. We then tested for an association in this simulated data. Power was reported as the proportion of simulations where the test returned a significant result. Mean power to detect associations was estimated at 13% at alpha = 0.05 (min = 0%, max = 96%), and '30/1019' tests had a power above 50% (see Methods). We found that the power was influenced by the number of times a mutation occurred, as well as the enrichment of the mutation causal channel in the signature compared with average in the cancer type (Multiple Regression, $P < 2E-16$ both variables).

**Mutational processes shape driver mutation landscape**. There were 43 significant correlations between signature activity and driver mutations (Mann–Whitney $U$-test, false discovery rate = 0.05; one-sided test), out of 1019 triplets of specific mutations in individual driver genes, mutational signatures and cancer types tested. Three of the associations involved signatures linked to extrinsic mutational processes (i.e., mutagens), 30 involved signatures linked to intrinsic mutational processes and 10 involved signatures with no known aetiology (Supplementary Data 1 for the full list of associations).

Of the associations involving signatures linked to extrinsic mutational processes, signature 4, linked to smoking, was associated with *KRAS* G12C (CCA > CAA) in lung adenocarcinoma (Fig. 2a) and with *CTNNB1* D32Y (TCC > TAC) in liver cancer. Signature 24, linked to aflatoxin, was associated with *TP53* R249S (GCC > GAC) mutations in liver cancer.

There were multiple associations involving signatures linked to intrinsic mutational processes. APOBEC activity (Signatures 2 and 13) had 11 associations. Remarkably, *PIK3CA* E542K (TCA > TTA) and E545K (TCA > TTA) were associated with these signatures across five cancer types, accounting for 82% (9/11) of all APOBEC associations (Fig. 2b). In addition, *PIK3CA* E453K (TCT > TTT) was associated with an APOBEC signature in breast cancer.

DNA mismatch repair (MMR)-linked signatures (signatures 6, 15, 20 and 26) showed nine positive associations across four cancer types (stomach, colorectum, uterine carcinoma and glioma low grade). Of these associations, *PIK3CA* H1047R (ATG > ACG) occurred twice. *FBXW7* R465C (GCG > GTG), was associated with MMR signatures in both colorectum and stomach cancer (Fig. 2c). *KRAS* G12D (ACC > ATC) and *KRAS* G13D (GCC > GTC) were associated with MMR signatures in uterine carcinoma and stomach cancer, respectively (Fig. 2d). These results suggest an important role for MMR defects shaping the driver mutation spectrum of common cancers, and illustrate the likely sequence of events (early MMR-linked mutational processes relative to driver mutation acquisition) in some cancers with these defects.

Nine associations with deficiency in DNA-proofreading (signature 10) were seen in uterine carcinoma and colorectum. *PTEN* R130Q (TCG > TTG) was associated with this signature in both colorectum and uterine carcinoma (Fig. 2e). 2/11 positive associations involved stop-gain mutations in the *APC* gene, one in colorectum and one in uterine carcinoma. Therefore it appears that *POLE* defects may cause characteristic driver lesions in these cancer types.

Six of the associations involved signatures that are known to correlate with age at diagnosis[17]. Of particular note, signature 1 was associated with *APC* R213X (ACG > ATG) in colorectum (Fig. 2f). This result in particular highlights the important role of ageing-related processes in cancer development.

Our test for correlation between mutational processes and driver mutations focussed on processes which exhibit higher

activity of the causal channel. This reduces the overall number of tests and increases the power to detect putative associations. However, to probe whether mutational processes and driver mutation acquisition are correlated in general, we repeated the analysis above without restricting the tests to signatures where the frequency of the causal channel was above average in the cancer type. An enrichment for positive associations between driver mutations and signatures where the underlying process has a higher than average activity of the causal channel would be indicative of a mechanistic relationship. Indeed, we found that 24 out of 37 significant associations had higher than average channel activity, compared to only 13 cases where the causal channel was lower than average ($P$ = 5.5E-5; Fisher's Exact Test; Supplementary Fig. 4; Supplementary Data 2 for the full list of associations), supporting the notion that the respective mutational processes are responsible for the driver mutation. However, as our analysis is correlative, we cannot entirely rule out the possibility of other explanations for these associations. Despite this, the results above support a model whereby mutational processes play an important role in determining driver mutation spectrum.

We note that there are cases of a common driver mutation in a cancer, which do not match the dominant signature in that cancer type. For instance *BRAF* V600E (GTG > GAG) is not explained by signature 7, linked to ultraviolet exposure. Neither is *BRAF* V600E (GTG > GAG) explained by the dominant MMR-linked signature 6 in colorectal cancer. Similarly, *PTEN* R130G (ACG > AGG) is very common ( > 86% of samples) in uterine carcinoma but is not explained by signature 6, which is also dominant in this cancer. These cases point toward a key role for selection in addition to mutation in determining driver mutation incidence. In the next section, we turn to explore the role of selection in greater detail.

**Detecting differential selection**. Driver mutations are recurrent in cancer because they experience positive selection. Consequently, the frequency that a particular driver is observed across cancers is a function both of the mutational likelihood of it occurring in the first place, and also the selective advantage that the mutation confers. Accordingly, the selective difference between the mutations can be inferred by normalising the observed frequency of the mutations across cancers by their underlying mutational likelihood (see Methods for mathematical framework; Fig. 1b for a graphical representation). With this logic as our foundation, we therefore aimed to quantify the differences in selective advantage between (typically) mutually exclusive driver mutations, using the results from the first part of this manuscript to normalise for mutational likelihood.

To test for differential selection between two related mutations in a cancer type (e.g., mutation of different residues of the same driver gene), we calculated the frequency of each mutation and their relative likelihoods of occurrence, inferred from the mutational process exposures (as per the analysis above). We then used the Poisson binomial test to examine the null hypothesis that the mutation counts were explained solely by their relative mutational likelihood of occurrence (see Methods). We explored potential differential selection among the most common driver mutations ( > 1% of non-synonymous mutations) in nine genes: *KRAS, BRAF, NRAS, IDH1, IDH2, TP53, PIK3CA, SMAD4* and *CTNNB1* (Supplementary Data 3) in individual cancer types. We conducted pairwise tests among the mutations from each gene in each cancer type where the common mutations in the gene occurred at least 10 times.

**Differential selection within driver genes**. Differential selection between driver genes was common. In total, 19% (655/3476) of

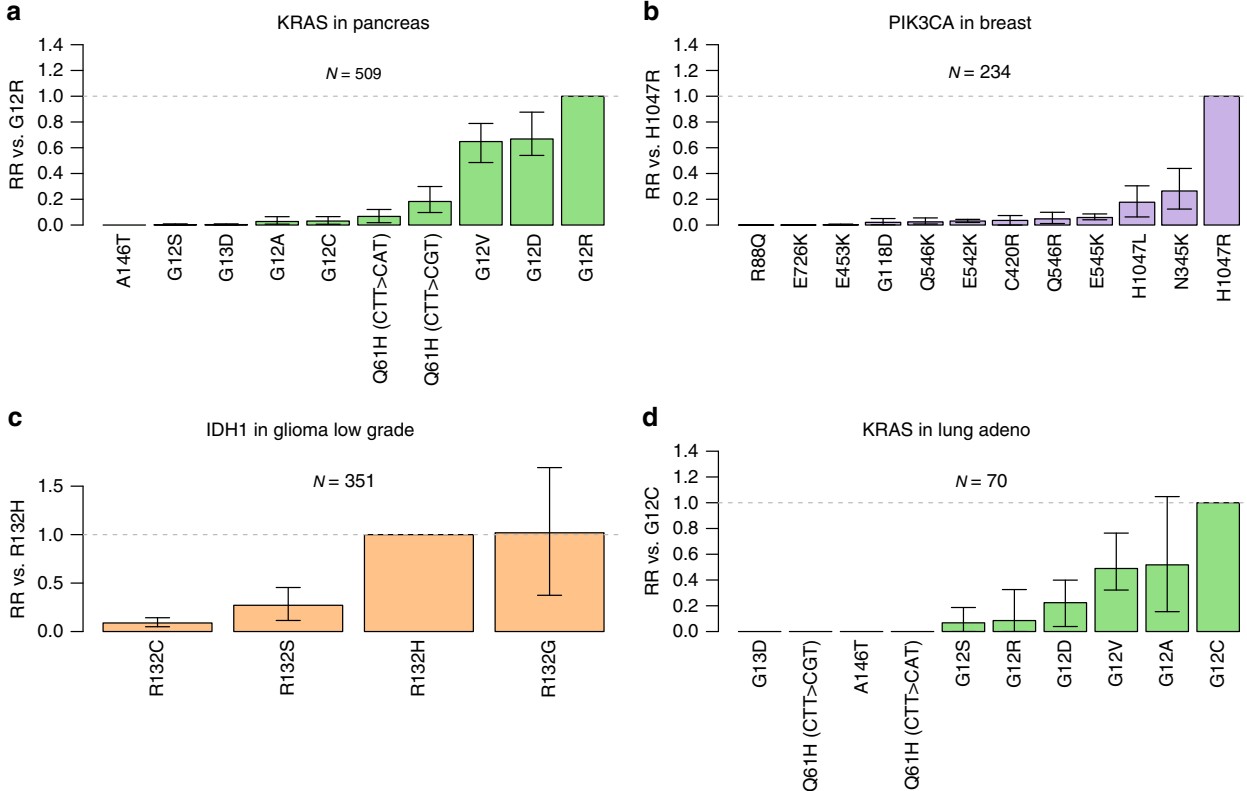

**Fig. 3** Evidence for differential selection between mutations in the same driver gene. **a** Modelled relative risk (RR) of frequent KRAS mutations in pancreatic cancer compared with *KRAS G12R*. For each mutation, the maximum likelihood estimate of relative risk compared to *KRAS G12R* is shown. **b**, **c** and **d** illustrate modelled relative risk for respectively, *PIK3CA* mutations in breast cancer, *IDH1* mutations in glioma low grade and *KRAS* mutations in lung adenocarcinoma. Grey dashed line indicates relative risk of one. Error bars represent 95% confidence intervals obtained by bootstrapping across 100 iterations. *N* indicates the number of samples used for the analysis within each cancer type

pairwise comparisons between mutational likelihood corrected frequency of mutations of different residues in the same gene in individual cancer types returned a significant result (Binomial Test, FDR = 0.05, Supplementary Figs. 5–15). All nine genes examined had at least one pair of mutations that occurred at frequencies inconsistent with the underlying mutational likelihood (Supplementary Data 4).

Among the most highly significant results, *KRAS G12R* appeared more strongly selected than other *KRAS* mutations, including *KRAS G12C* and *G13D*, in pancreatic cancer (Fig. 3a, Supplementary Fig. 7), as did *BRAF V600E* compared with other *BRAF* common mutations, including *BRAF K601E*, in thyroid, melanoma and colorectal cancer. Also highly significant was apparent preferential selection for *PIK3CA H1047R* compared with multiple *PIK3CA* mutations, including *PIK3CA E545K* and *E542K*, in breast cancer (Fig. 3b, Supplementary Fig. 9), and for *NRAS Q61K* and *Q61R* above *NRAS G12D* and *G13D* in melanoma. These results suggest that there are strong selective differences among important driver mutations in the same gene in these cancer types.

A number of the results are of potential therapeutic interest. For example, we found evidence that *IDH1 R132H* is selected more strongly than *IDH1 R132C* in low-grade glioma (Fig. 3c, Supplementary Fig. 14) and glioblastoma. This is of particular interest given the potential specificity of therapeutic small molecular inhibitors that target *IDH1* and *IDH2* mutations[18].

*KRAS G12C*, which was found to be associated with smoking-associated signature 4 in lung adenocarcinoma (see above), also appears more strongly selected than other *KRAS* mutations (including *G12D*, *G12R* and *G13D*) in this cancer type (Fig. 3d,

Supplementary Fig. 7). Thus, it appears that the high frequency of this *KRAS* mutation compared with others in lung adenocarcinoma is, potentially, owing to both smoking-associated mutational processes and the intrinsic selective advantage of the mutation.

Interestingly, the relative selective advantages of particular pathogenic mutations in each gene were broadly consistent across cancer types. Specifically, there were only 7/118 cases of differentially selected mutations where a mutation appeared selected more strongly than another in the same gene in one cancer type, but less strongly in another cancer type. As our method controls for differences in mutational process activity between cancer types, these results provide evidence to support the hypothesis that the mechanisms that underpin the selective advantage caused by a specific driver mutation are broadly uniform across tissue types.

**Differential selection between driver genes**. We next used the same methodology to investigate differential selection between mutations within and between small sets of genes that typically show mutually exclusive mutation patterns. We considered the common driver mutations in three sets of functionally related genes: *KRAS*, *BRAF* and *NRAS*; *APC* and *CTNNB1*; and *IDH1* and *IDH2*.

There was evidence of greater selective differences between genes than between different residues within a gene. 12% (306/2,541 pairwise comparisons) of tests were significant for mutations within a gene, whereas 28% (841/2995) were significant for mutations in different genes (Supplementary

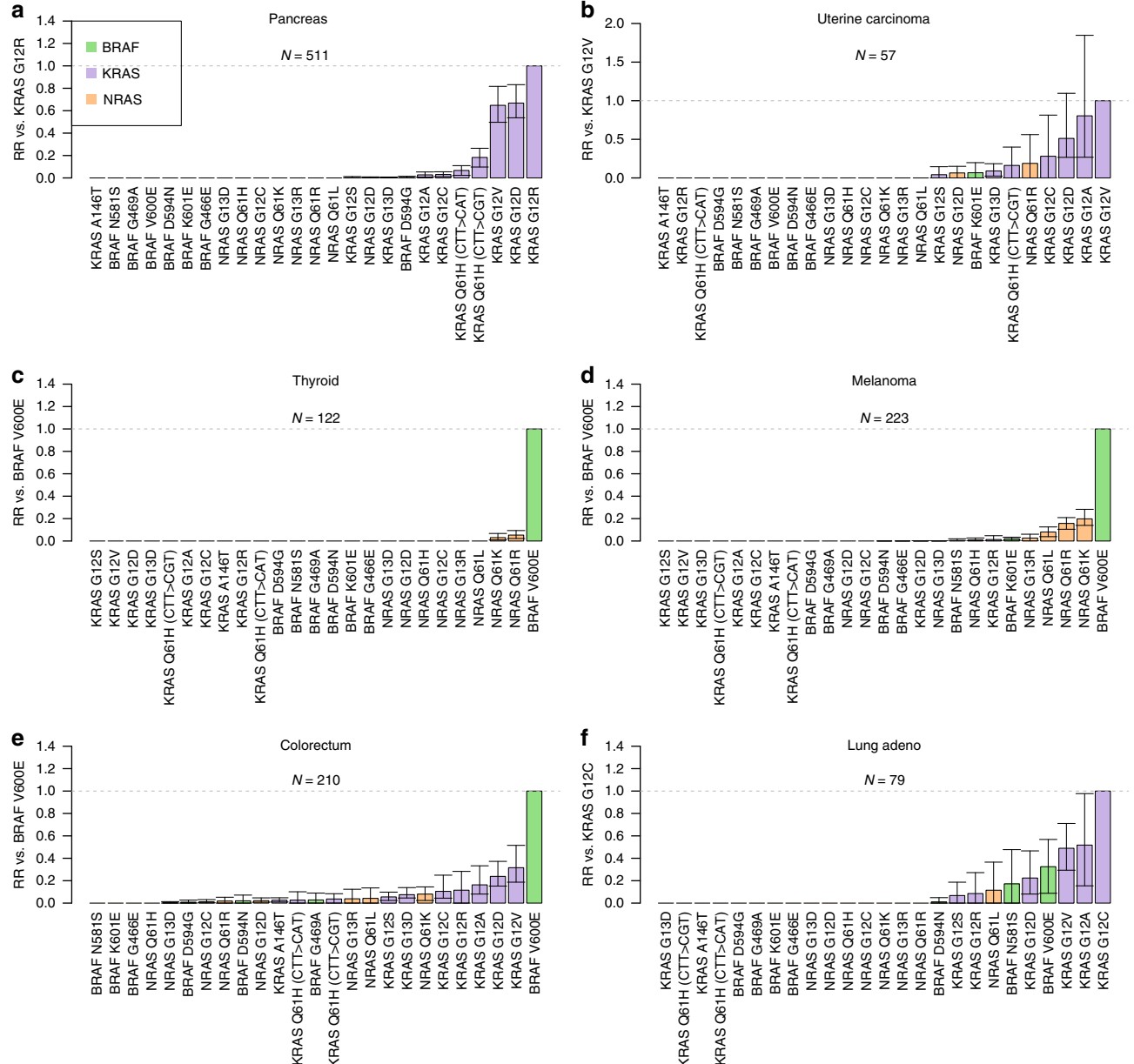

**Fig. 4** Evidence for differential selection between mutations in *KRAS*, *BRAF* and *NRAS*. Bar plots show modelled relative risk of *KRAS*, *BRAF*, and *NRAS* mutations (compared with a reference mutation). **a** Modelled relative risk of *KRAS*, *BRAF* and *NRAS*, mutations compared to *KRAS G12R* in pancreatic cancer. **b** As above, with comparison to *KRAS G12V* mutations in uterine carcinoma. **c** As above, with comparison to *BRAF V600E* mutations in thyroid cancer. **d** As above, with comparison to *BRAF V600E* mutations in melanoma. **e** As above, with comparison to *BRAF V600E* in colorectum. **f** As above, with comparison to *KRAS G12C* in lung adenocarcinoma. Confidence intervals obtained by bootstrapping across 100 iterations. Error bars represent 95% confidence intervals obtained by bootstrapping across 100 iterations. *N* indicates total number of samples used for the analysis within each cancer type

Figs. 16–20; Supplementary Data 5). Furthermore, for two of the mutation sets–*KRAS*, *BRAF* and *NRAS* (Supplementary Fig. 21); and *APC* and *CTNNB1* (Supplementary Fig. 22)—there was significant heterogeneity across cancer types in terms of the number of mutations in each gene with evidence of preferential selection (selection above at least one other mutation in the set) (Fisher test, $q = 2.2 \times 10^{-3}$, $7.2 \times 10^{-7}$, respectively), supporting a model where gene-specific effects on selection vary across cancer types.

Among *KRAS*, *BRAF* and *NRAS* mutations, only particular *KRAS* mutations showed evidence of preferential selection over mutations in other genes in pancreatic cancer and uterine carcinoma (Fig. 4a, b), whereas preferential selection across genes

was predominantly in favour of *BRAF* and *NRAS* mutations in melanoma and thyroid cancer (Fig. 4c, d). Illustrating this, BRAF V600E and NRAS Q61R appeared to be selected more strongly than KRAS G12D in melanoma and thyroid cancer, but more weakly than this mutation in pancreatic cancer. Other cancer types showed a range of patterns of differential selection for these three genes (Fig. 4e, f, Supplementary Figs. 16–18).

When *APC* and *CTNNB1* mutations were compared, there was evidence for selection of *CTNNB1* mutations over common *APC* mutations in each of liver cancer, uterine carcinoma, prostate cancer and colorectal cancer (Fig. 5a–c Supplementary Fig. 19). Interestingly however, evidence for selection of *APC* mutations above *CTNNB1* mutations was found in colorectal cancer only

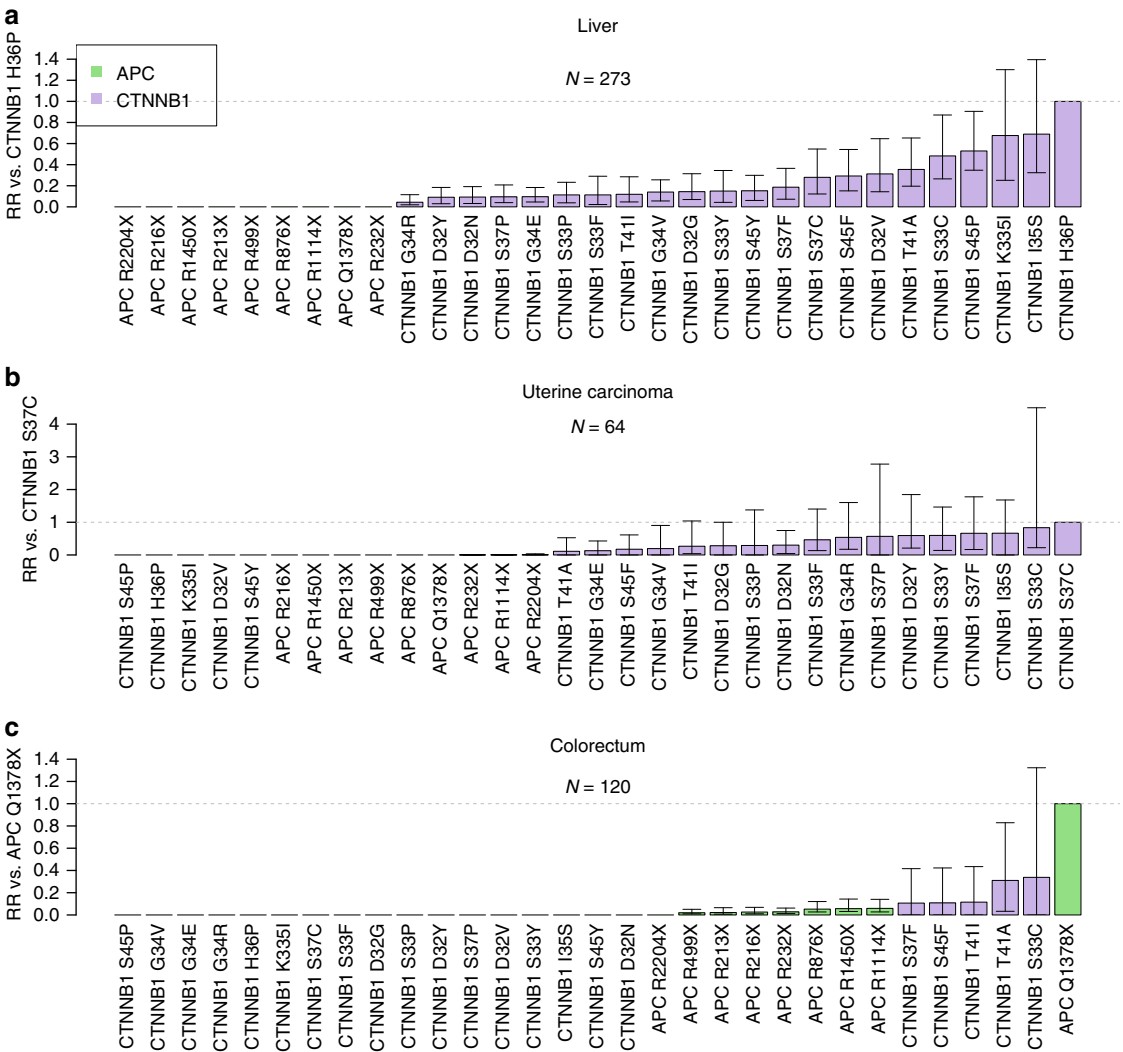

**Fig. 5** Evidence for differential selection between mutations in *APC* and *CTNNB1*. **a** liver cancer, **b** uterine carcinoma and **c** colorectum. Confidence intervals obtained by bootstrapping across 100 iterations. Error bars represent 95% confidence intervals obtained by bootstrapping across 100 iterations. *N* indicates total number of samples used for the analysis within each cancer type

(Fig. 5c). We note that this mutation, *APC Q1378X*, falls within the functionally defined mutation cluster region of the gene[19].

Among *IDH1* and *IDH2* mutations, we found preferential selection for *IDH1 R132H* (glioma low grade, AML and glioblastoma), *IDH1 R132C* (glioma low grade, AML, liver cancer and melanoma), *IDH1 R132G* and *IDH1 R132S* (glioma low grade) above common *IDH2* mutations, as well as preferential selection for *IDH2 R172K* above *IDH1 R132C* in glioma low grade (Supplementary Fig. 20).

Taken together these results inform our understanding of the selective landscape experienced by driver genes, and its similarities and differences between cancer types. Across the cancer types and mutation sets considered there was a positive correlation (correlation coefficient > 0) between mutation frequency and mutation probability in 58/76 cases (Supplementary Figs. 23–38). Among these cases on average 20% (mean $R^2$) of variation in frequency between related mutations within the cancer type is explained by variation in mutation probabilities. This suggests an important role for selective differences in explaining this variation. These results suggest that both intra-gene and inter-gene effects contribute to differential selection, with inter-gene but not intra-gene effects varying across cancer types.

## Discussion

Here, we have demonstrated correlations between mutational processes and key driver mutations across cancer types, and highlighted the possibility that these correlations may actually be the result of the mutational process causing specific driver mutations. Moreover, by normalising for mutational likelihood we have quantified relative selective differences between related key driver mutations across cancer types, which sheds light on the selective landscape constraining cancer evolution.

Many of the associations between mutational processes and driver mutations presented here are novel to the best of our knowledge, and warrant further molecular investigation to explore causality. Our analysis suggests that an ageing-associated process (signature 1) may cause initiating events in colorectal cancer because of the implied role of the process in causing *APC R213X* 'gatekeeping' mutation in colorectal cancer[16,19]—suggesting a sometimes critical role of 'bad luck' in this cancer type[20].

Previous work by McGranahan et al.[13] examined the relationship between APOBEC-associated mutational processes (signatures 2 and 13) and driver mutations and found that clonal non-synonymous mutations in driver genes occur in an APOBEC context in bladder cancer. They also described subclonal mutations in driver genes in an APOBEC context in bladder, breast,

head and neck, and lung cancers (cervical cancer was not considered). Supporting their findings, we detected associations with APOBEC in bladder cancer and breast cancer, and to a lesser extent in head and neck, lung squamous and cervical cancer. Notably, we report novel associations between APOBEC activity and *ERBB2 S310F* mutations in bladder cancer. Our findings support the impression of a pervasive effect of APOBEC activity on driver mutation spectra in human cancers. Some associations we describe have been reported previously, notably the association between pack years of smoking and the *KRAS G12C* mutation in lung adenocarcinoma where the connection between the causal channel of this mutation (C > A in a CCA context) and the general tendency for tobacco carcinogens to cause transversions is well known[21,22].

Remarkably, 14/43 associations between mutational signatures and driver mutations involved *PIK3CA* mutations, and most of these associations involved signatures linked to APOBEC, which tends to occur later in carcinogenesis[13]. Thus, late arising APOBEC-linked mutational processes can still have important influences on the driver mutation spectrum. Recent results showing that *PIK3CA* mutations are often subclonal[13] support this interpretation.

Through normalising for mutational likelihood, we have also been able to quantify the relative contribution of clonal selection, over and above mutational likelihood, in determining driver mutation spectra across cancers. We found evidence for widespread differences in selective effects between mutations in the same gene and related genes, and moreover, that these differences appear to vary across cancer types. These results confirm that not all driver mutations have the same selective effects, and instead exist on a spectrum of selective potency. Both mutational likelihood and selective difference strongly contribute to the occurrence of specific driver mutations in cancers.

Previous methods have taken into account the variability of mutation rates between mutation types to infer selection[23–25]. Through the use of the dN/dS measure (the ratio of non-synonymous mutations to synonymous mutations within a gene) two of these studies have gone some way to quantifying the effects of selection experience by individual genes[24,25], beyond a binary distinction between driver genes and passenger genes. However, all these studies quantify selection at the level of genes, rather than the level of individual mutations. To the best of our knowledge, this study is the first to take into account the variation in background mutation rates to quantify selection experienced by individual nucleotide driver mutations in a pan-cancer analysis.

The exact mechanistic cause of the selective differences we have identified will be an important area for future work. The differences we identified could reflect variation in the potential of the mutations in question to initiate disease, or alternately variation in the growth advantages conferred by these cells in established tumours. Interestingly, if there are differences in on-going growth advantages, then our data suggest that the forces of selection acting in tumours are often insufficient (or have insufficient temporal opportunity[26]) to displace sub-optimal mutations, as less highly selected mutations remain detectable. For a limited number of driver genes, there is evidence to suggest that specific mutations correlate with disease outcomes[27,28]. Further work is needed to clarify to whether and to what extent the selective differences indicated here have prognostic and therapeutic implications.

In lung cancer, the *KRAS G12C* mutation provides a striking example of the potential for 'alignment' of mutation and selection: the likelihood of the *KRAS G12C* mutation is increased by smoking, but in addition it is also selectively advantageous above other common *KRAS* mutations in the disease. The same is also true for *PIK3CA H1047R* mutations in stomach cancer, wherein

MMR-associated processes increase the likelihood of the driver mutation, which is then subsequently strongly selected.

In this study, we have analysed the influence of mutation and selection on single-nucleotide alteration frequencies. In theory a similar analysis could be carried out for copy number alterations or methylation changes. At present, the limited knowledge of pan-cancer mutational signatures involving these types of change make this challenging. However, the expected publication of greater numbers of whole-genome sequencing and methylation studies could make this possible in the near future.

There are caveats to this analysis. First, we have used data from a number of sources, which may vary in terms of quality, depth of coverage and the pipeline used to call mutations. Second, we have relied on the assignment of signatures to individual samples and we note that some samples have relatively few mutations, making this assignment less accurate. Relatedly, in some cancer types, there are other active signatures that were not considered in this study. Where other signatures are present, the regression method used here can only approximate the signature contributions. Third, some mutational signatures are similar to each other in composition, making it difficult to determine whether mutations are generated by one or more independent processes. We rely on assumptions of uniformity of a mutational process across the genomic loci considered, and over time. The combined consequences of these caveats could lead to false positives or false negatives in the associations between mutational signatures and driver mutations and the inference of differential selection. We note that based on data from MutSigCV[23], genes similar to *KRAS*, *BRAF,* and *NRAS*, in replication timing and expression have similar background mutation rates, as do genes similar to *APC* and *CTNNB1*. The background mutation rates of genes similar to *IDH1* and *IDH2* show a greater difference, but one that is still smaller than selective differences inferred between these genes. Finally, causal links between driver mutations and mutational processes are one explanation for the associations presented here, but other explanations cannot be ruled out from these data alone.

In summary, our framework quantifies the combined influence of both mutation and selection on shaping a cancer's driver mutation complement—and importantly emphasises that neither evolutionary force alone provides a sufficient explanation of the observed mutation distribution. In colon cancer, for example, *BRAF* mutations (that are relatively uncommon) are mutationally unlikely, but are strongly selected. By contrast, *KRAS* drivers (that are more common), are mutationally much more likely, but are less highly selected. Our data also offer an explanation for the high frequency of driver *APC* mutations and relative paucity of driver *CTNNB1* mutations in the colon: *APC* mutations can be both more strongly selected and more mutationally likely than *CTNNB1* mutations.

Overall, our results begin to quantitatively delineate the distinct contributions of mutation and selection in shaping the spectra of driver mutations in the cancer genome.

## Methods

**Sample-specific mutation collection**. Only mutations on canonical nuclear chromosomes were considered. For ICGC data, mutations labelled as 'single-base substitution' in the simple somatic mutation files were considered for further analysis. For TCGA data, only mutations labelled as 'SNP' in the mutation annotation files were considered.

From these lists, non-synonymous mutations in driver genes were extracted. Driver genes definitions are as stated below. After filtering for drivers, these mutations were re-annotated using Annovar[29]. We included mutations labelled as 'non-synonymous SNV', 'stopgain' or 'stoploss' in a driver gene in the annotation by Annovar.

**Definition of driver genes**. Driver genes were defined using a recent study by Vogelstein et al[16]. The list of genes is given in Supplementary Data 6.

**Signature exposures for each sample in each cancer type**. The 96-channel context of each SNV was imputed using the R package 'SomaticSignatures'[30], and the total number of SNVs in each of the 96 channels was calculated for each sample. Non-synonymous mutations in driver genes were excluded. Mutational signatures were obtained from the Wellcome Trust Sanger Institute (http://cancer.sanger.ac.uk/cosmic/signatures) in April 2016. Information on the presence/absence of these signatures in individual cancer types was obtained from the same source. For whole-exome data, signatures were re-scaled to the trinucleotide frequencies of the exome. Non-negative least squares regression, implemented in the R package 'nnls'[31], was used to assign a process activity for each signature reported as being present in the applicable cancer type. Signature activities were normalised for each sample to calculate the signature exposures. We note that analyses based on absolute signature activity gave broadly consistent results.

**Required mutations for signature assignment**. By treating each of the 30 signatures as a multinomial probability distribution, we simulated data sets from each signature with n total informative mutations ($1 < n < 96$). For each signature, for each value of $n$, we applied non-negative least squares regression to the simulated data to assign weights to the true generating signature and a set of 14 randomly chosen other signatures. We classified the regression as successful when over 50% of the regression weights were assigned to the true signature. We chose to use 15 possible generating signatures as this was above the maximum number of signatures identified in any individual cancer type. For each signature, for each number $n$ of informative mutations, we calculated the proportion of simulated data sets where the regression was successful. We found that 20 mutations gave an average classification accuracy of 80% across signatures. As a result, we chose to use a cutoff of 20 mutations to strike a balance between including as many tumour samples as possible while still maintaining reasonable accuracy of signature assignment. We repeated the analysis of associations between driver mutations and mutational signatures using a cutoff of 50 mutations per sample for comparison. This analysis recovered 41 associations, of which 37 were also found using the 20 mutation cutoff.

**Power calculations**. We sought to test the power to detect an association between mutation M and the signature A in cancer type C, where M occurred m times in C. We considered a simple model of cancer initiation, where M is one of a set of mutations R of size $|R| = n$, one of which is required for cancer initiation. For these purposes we assumed n = 10.

For each random iteration of the power model we randomly selected causal channels out of 96 possibilities of the nine other mutations in R. We identified the signature exposures of each sample in C. By treating the signatures as multinomial probability distributions, we then calculated the per sample probabilities that mutation M occurred rather than any of the nine mutations in each sample. Based on these probabilities we randomly selected m samples to bear the mutation M. We then applied the Mann–Whitney U-test described above.

The power was calculated as the proportion of iterations where the p-value in the test was less than the quoted value of alpha.

Out of 1019 triplets tested where a signature represented a fold increase in the causal channel of a recurrent driver mutation in a cancer type, relatively few significant associations (43) were found. The low number of associations can be partly explained by the low average power. Even if associations were genuinely present in every case, the expected number of significant tests was 130 based on the estimated power. Part of the reason for this is the technical challenges inherent in deconvolving mutational signature intensities. Timing mismatches between the activity of a mutational signature and the window of selection for a driver mutation probably also contribute to the low numbers of associations.

**Mathematical framework**. Following Tomassetti[32], we make the simplifying assumption that a mutation $M_i$ occurs at a constant low rate $u_i$ per year, $u_i << 1$. Suppose that after the occurrence of the mutation sequence $R = < M_1, M_2, \ldots, M_n >$ cancer occurs with constant rate $\lambda << 1$. Then by well-known results[32,33] the probability of cancer incidence at time t is given by:

$$I(t) = \frac{(u_1 u_2 \ldots u_n \lambda t^n)}{n!}$$

Extending this framework to take into account cancer causation by multiple sequences of mutations $S_j = < M_1(j), \ldots, M_n(j) >$, with rate $\lambda_j$. Cancer incidence at time $t$ is given by

$$I(t) = \frac{\sum_j \left( u_1(j) \ldots u_n(j) \lambda_j t^n \right)}{n!}$$

The above closely follows[32].

Definition—Mutations $M_1$ and $M_2$ are similar with relative risk $r_{1,2}$ if they satisfy the following property: a mutation sequence $S_j$ containing $M_2$ causes cancer with rate $\lambda_j$, just if the mutation sequence $S_k$, that results from substituting $M_1$ for $M_2$ in $S_j$, causes cancer with rate $r_{1,2}\lambda_j$.

We note that by this definition all mutations are similar to themselves with relative risk 1.

Now consider two mutations $M_1$ and $M_2$, that are similar with relative risk $r_{1,2}$. Then the probability that $M_1$ occurs in a cancer sample, given that either $M_1$ or $M_2$ occurs is given by:

$$P(M_1 | M_1 \cup M_2) = \frac{u_1 r_{1,2}}{(u_1 r_{1,2} + u_2)}$$

**Test for differential selection between mutations**. Given two mutations $M_1$ and $M_2$ in a cancer type C, we identified the samples in which exactly one of the two mutations occurred, and no other mutation in the set of mutations under consideration for differential selection occurred. For each such sample $S_i$, we calculated $p_{i1}$ and $p_{i2}$, the probability of the causal channels of $M_1$ and $M_2$, respectively, occurring among the 96 mutation types, based on mutational signature exposures for the sample.

We used a Poisson binomial test to test whether the frequencies, $m_1$ and $m_2$, of $M_1$ and $M_2$ were consistent with their relative probabilities of occurrence across the samples. Specifically, we modelled $m_1$ with the random variable $X$, where:

$$X \sim \text{Poibin}(\mathbf{q})$$

$$q_i = \frac{p_{i1}}{(p_{i1} + p_{i2})}$$

We then used a two-tailed test to test whether the observed data differed from the predicted distribution.

**Modelled relative risk**. Given two mutations $M_1$ and $M_2$, with probabilities of occurrence $p_{i1}$ and $p_{i2}$ in sample $S_i$, we model the probability $q_i$, that $M_1$ is present given that either $M_1$ or $M_2$ is present by the following formula:

$$q_i = \frac{(r_{1,2} * p_{i1})}{(r_{1,2} * p_{i1} + p_{i2})}$$

where the parameter $r_{1,2}$ is the relative risk of mutation $M_1$ compared with mutation $M_2$.

Defining $I_1$ and $I_2$ as the sets of sample numbers where $M_1$ and $M_2$ occurred, respectively, then the likelihood of the data, $L$, is given by:

$$L = \prod_{(i \in I_1)} q_i * \prod_{(i \in I_2)} (1 - q_i)$$

We used numerical methods to find the maximum likelihood estimate of $r_{1,2}$ for each pair of mutations in each tumour type, based on this formula. Bootstrapping with 100 iterations was used to find approximate confidence intervals around these estimates.

**Comparison of genomic and exonic mutation distributions**. Our study used a combination of whole-genome sequencing (WGS) and whole-exome sequencing (WXS) data. The 1441 whole WGS samples were distributed predominantly across 8/22 cancer types. In total, five associations between mutational signatures and driver mutations were identified across these eight cancer types. All five of these associations were in Liver cancer, where 27% of samples (305/1110) were WGS samples.

To assess the effect of using both whole-genome and whole-exome samples on our analysis, we analysed the effect on our results of replacing the WGS data with only the exonic subset of mutations. We recovered 41/43 associations between driver mutations and mutational signatures and found no new associations, suggesting that using WGS data in addition to WXS has a limited effect on the analysis.

**Variation explained by mutation probability**. For each mutation, the probability of the mutation in each sample of a cancer type was calculated based on sample-specific mutational signature exposures. The mean probability across samples was found, as well as the number of times the mutation occurred. Linear regression was carried out to find the proportion of variance in mutation frequencies across different mutations explained by variation in their probabilities.

**Data availability**. Mutation data (SNVs) were downloaded from the ICGC and TCGA data portals in May 2016 (https://dcc.icgc.org/; https://tcga-data.nci.nih.gov/docs/publications/tcga/). We excluded data sets aligned to a reference genome other than hg19, and those with non-conforming formatting. Computer code is available on request.

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

## Acknowledgements

D.T. acknowledges funding from the EPSRC via CoMPLEX (grant #: EP/F500351/1). TG was funded by Cancer Research UK (grant #: A19771) and the Wellcome Trust (grant #: 202778/Z/16/Z). B.S.-B. received funding from Ludwig Cancer Research. I.P.M.T. acknowledges funding from Cancer Research UK A16459 and the EU EVOCAN ERC award.

## Author contributions

D.T., I.P.M.T., B.S.-B. and T.A.G. jointly conceived and designed the study. D.T. performed all bioinformatic and statistical analysis. All authors analysed the data. D.T., I.P.M.T., B.S.-B. and T.A.G. wrote the paper. T.A.G. supervised the study.

## Additional information

**Competing interests:** The authors declare no competing interests.

