## [Peer Review File · Nature Communications]

Reviewers' comments:

Reviewer #1 (Remarks to the Author):

Temko et al present an in silico investigation exploring the putative relationship between mutational processes and driver mutation acquisition across multiple cancer types. In principle, I think this is quite an interesting analysis.

I would however, suggest some improvements:

- in the writing of the manuscript
- in description of the methods in order to ensure that all is absolutely transparent
- tidying up the signature extractions and assignments

Detailed critique:

INTRODUCTION

First para ok.

Second para ok.

Third para – this is an important para:

– improve description in this first sentence. It is apparent after reading the whole manuscript what you mean in this sentence but to a naïve reader, this sentence is entirely nonsensical. Keep the choice of words simple here and take the time to explain what you mean.

- Some punctuation missing,

o Traditionally (comma)

o Here (comma)

- Some word choices could be improved:

o Here we present not analyse

o We also explore not analyse

RESULTS

Testing for mutational signature and driver associations

• Para1: mutational signatures do not act on anything. Mutational processes are active. The signatures are the imprints left behind by those processes. I would change “acts” to “is active” or revamp the sentence completely.

• This sentence: “Where the activity of a mutational signature was significantly higher in cancers with a mutation of interest compared to those without, we considered it prima facie evidence of a causative relationship between the mutational signature activity and the presence of the driver mutation”.

This is the crux of this manuscript and is quite an exciting way to think but must be toned down.

The reason is that it is never causative and only ever associative. Furthermore, there are signatures that are associated with each other, so you may be seeing a relationship that is secondary... so.. I would be careful about how this is worded.

• The mutational signature analysis: How did you perform the extraction? Did you use SomaticSignatures R package using the 30 COSMIC signatures as an a priori, across the whole cohort of all those different tumor-types. (i.e. a global analysis) or did you do it tumor-type by tumor-type (local analysis). Doing a global analysis is a little dangerous for the next step - which is the assignments. It is well-known that the global analyses result in spilling of certain signatures into tissue-types that do not actually carry that mutational process biologically. This is a mathematical anomaly and various post-hoc (if slightly mad) filters have then been included to assign mutational signatures to individual samples in the past when global analyses have been performed. How did you deal with this? So for example smoking-associated Signature 4 is well-known to spill into all sorts of tumor-types (e.g. prostate) just because a particular C>A dominant

signature is present in that tumor-type (even though there is no smoking-associated process acting there. How did you exclude this problem? This is quite relevant to your analysis because your attribute a driver to a particular signature and imply causation. You need to make clear how you did your extraction and assignment.

- It is also well-known that some of the mutational signatures such as Sig 5 is notoriously close (in terms of profile/cluster centroid) to other signatures (Signatures 3 and 8), and also not necessarily easily distinguished from Signature 1. Indeed, in some tissue types, it remains controversial whether Signature 5 is actually separate from Signature 1 unless performed in a particular mathematical way (a priori forcing to find these as separate signatures). Indeed, you show in Fig S1 that the classification accuracy is not good for Signature 5 when compared to other signatures. My question is then this: How can you be sure that your signature assignments are accurate for each driver mutation? Can you demonstrate to the reader that you have given this some consideration, even if one cannot be absolutely sure about the assignments? For signatures 2 and 13, I am certain that your assignments are pretty good, but I am less certain about Sig 5.

Mutational signatures shape driver mutation landscape

- Typo. Aflatoxin

- Final para – this is sloppy use of terminology “linked to ageing”. Those signatures have been shown to be associated with age of diagnosis. That does not mean that they are pathognomonic of “ageing” (i.e. senescence). A correlation with age could be due to a number of factors – e.g. relative time or cell turnover – so to say that Signature 1 or 5 is linked to ageing is a step too far (though you are not the only ones to loosely use this description in this way).

Detecting differential selection

When you are looking for evidence of differential selection, you do a comparison to a meta-sample which you have defined as an average for the relevant cluster-group. But for rare mutational processes e.g. MMR in breast cancer, a sample will end up in a cluster but effectively be an outlier in that cluster. So, when you determine the likelihood for seeing something at a particular sequence context, your expectation is based on an average but your observed refers to a specific mutation in a specific sample (which could be an outlier). You could over-call some of these as significant. Can your background model be improved to take into account per sample differences.

Differential selection between pathogenic amino acid changes within a driver gene

- Second para needs to be improved in writing – this is quite an important para – and tie it up more precisely with your figures please. Fig 2B description comes after G13D but your figure is a comparison to G12D – it throws the reader. And I cannot see in Fig 2B how those show more strong selection than G13D at al.. Show the figure for G13D instead... seeing as you make the point of it in the manuscript. Or better still the KRAS G12C one... to highlight that all the symbols (same cluster) and mutation-type will be the same above the line – to emphasise that both mechanism and selection driving it. I do wonder whether there is a more intelligent way of demonstrating all this data – this is effectively a plot per comparison to each driver. The supplementary information is just hideous.

- Can you provide some sort of quantitative measure (with errors) of the effect of selection and present that instead (as a summarizing image/plot). That might give you more credence in demonstrating that you have a continuum of selective effects and help present the information more cohesively. Perhaps a heatmap where the 2D-distance from the grey dashed line is given for

each mutation type in each gene. You will have 8 plots (one for each gene). Just a suggestion – the point is I think to think of a better way to present this information.

- Last para: I do not think that this is a fair comparison, across tissue-types because they will have very different prevalences of signatures.

Differential selection between mutationally exclusive driver genes

So I get this but it is not easy to follow this section and it is not easy to walk away with some form of over-arching sense of the message here without feeling like there has been only a token correction for mutational process. If you present data against a particular mutation (e.g. E545K), do you do it for the specific mutations in each cluster, or to all E545K mutations in samples that belong to any cluster for that tumor-type? I can't tell at the moment.

Figure 1

- a) Signature 5 H matrix to generate this must be contaminated with signature 1 very heavily. This does not look like the signature extraction and assignment has been very accurate.
- b) I do not understand what this figure is trying to show... nor why it would be relevant. If I understand it correctly, it's simply showing the averaged 96-channel mutation profile across all signatures that are active for that tumor-type. It's very distracting and unhelpful.
- c) Asterisk not explained. What is p-value. Please name statistic used to do the comparison (note: non-normal distributions).

Figure 2: OK

Figure 3 and 4 are ok but I think you could condense the information a little more, it's a lot of disparate plots and the message is slightly lost.

SUPPLEMENTARY

- Fig S14 missing
- Description of Figure S15 and S16 is required. No idea what I am looking at.
- Table S1, S3-S6 are OK
- Table S2 mutation frequency was calculated how? Need correction for frequency of triplets in genome/exome?

Reviewer #2 (Remarks to the Author):

Temko and colleagues present an analysis entitled "The effects of mutational processes and selection on driver mutations across cancer types". In this analysis the authors use publicly available whole exome and whole genome sequencing data from more than 10,000 tumour samples with respect to the relation of trinucleotide mutation spectra and recurrent coding base substitutions in known cancer driver genes.

The question to which extent the observed distributions of potential pathogenic hotspot mutations is caused by mutagenic processes alone, or due to positive selection, is important and has gathered considerable attention in the past years

The analysis presented consists of basically two parts. In the first part the authors assess the correlation of mutational signature activity and the presence of particular hotspots. This analysis is coupled with power calculations, which I highly commend.

There is however, a peculiarity in the sense that a positive correlation of a particular variant with the matched underlying mutational process generating this type of substitution rather suggests the absence of selection. Yet I guess what the authors are getting at is that the mutational processes are a strong determinant of the observed driver spectra. Wouldn't it be more informative, rather than testing for association to quantify the amount of explained variation, or an equivalent measure?

I also wondered how much of the association is explained by other effects such as transcriptional strand biases due to transcription couple repair, similarly replication strand biases for mismatch repair as well as age, gender and ethnicity. There could also possibly a higher order of the base context to be investigated.

Similarly, in my experience, the quantification of signature activity in individual samples can be very noisy with strong collinearity between inferred exposures. This uncertainty does not enter the test, except for the inclusion threshold of 20 mutations per sample, and I wondered whether this influences the outcomes. Related to this question is the choice of signatures - are the authors using any type of regularisation/model selection?

To demonstrate the validity of their test strategy it would be important to demonstrate the accuracy (QQ-plots) of their approach using simulations. Such simulations should recognise the aforementioned strand biases, transcription strength, replication timing, as well as the epigenetic associations with mutation rate that have been described by some of the authors before to make the background as realistic as possible. On genomes, this can be done using permutations within windows, for exomes one would need to establish a more elaborate way to keep the marginals fixed.

In the second part of the analysis the authors investigate the frequencies of different hotspots in the same driver gene and in mutually exclusive pairs of driver genes. The authors cluster samples based on signature activity and then test for differences in hotspot frequencies using a binomial test as a measure of selection.

While this analysis is interesting it appears to be a bit of a black box and it would again be important to demonstrate the accuracy of this approach using realistic simulations (see above). I would be worried that the clustering missed important information that could explain some of the observed variance and therefore leads to false positives.

While dealing with an interesting scientific questions, in the absence of a systematic evaluation of the tests performed in this analysis I find it difficult to judge the reliability of the results.

Reviewer #3 (Remarks to the Author):

Temko and colleagues used public cancer sequencing data to infer the independent effects of

mutation and selection driver mutation complement. Their results suggest that selection is the key determinant of which driver alterations dominate in different cancer types. While an interesting topic, the manuscript lacks a coherent structure and this reviewer found it a little unclear, making it difficult to draw meaningful conclusions and provide a comprehensive review.

A number of important points must be addressed before this article can be considered for publication:

1. The authors use both whole genome and predominantly whole exome sequencing analysis. Given that these samples will contribute vastly different numbers of mutations, the authors must demonstrate that this does not affect their results. For instance, if they simply use exome sequencing data, are the results concordant?
2. A central part of the analysis assumes accurate classification of mutational signatures. However, the methods are not entirely clear in this regard. A number of publications exist to identify mutational signatures and assign known signatures to samples – e.g. Alexandrov 2014, Rosenthal, 2016, Kim, 2016. Rather than using a published method to assign mutations to signatures the authors use non-negative least squares regression, implemented in R. The authors must demonstrate that their method is at least as good as published methods. Ideally, the authors should also implement published methods and demonstrate that their results are robust.
3. Associating a single point mutation to a particular signature can be a little problematic. The authors should discuss the limitations of this approach, and the fact that certain signatures are far less specific than others. For instance, while it may be relatively easy to assign a particular driver mutation to an APOBEC (Signature 2 or 13) context, discriminating between a C>T mutation at a CpG site in Signature 5 from one in Signature 1 is rather trickier.
4. This manuscript would benefit from an additional Figure, preferably at the start, outlining the key questions that are being addressed and the methods used to address these. Such a figure may provide a point of reference for the reader, and, importantly, make the paper considerably easier to read. Indeed, without such a figure, this reviewer struggled to fully grasp the key biological questions addressed in the manuscript.
5. The authors normalize the signature activity to one for each sample, to calculate signature exposures. Such an approach may result in certain signatures 'drowning' out other signatures. For instance, elevated APOBEC mutational signature may result in a, relatively speaking, lower Signature 1, related to aging. However, this does not mean that the mutational process associated with aging necessarily contributed less to tumor development than in another tumor that does not exhibit APOBEC mutations. Thus, the authors should consider utilizing the total number of mutations associated with a signature rather than proportion of each signature present.
6. The authors suggest 20 mutations give 'an average classification accuracy of 80% across signatures'. However, given that only 50% of the regression weights were required to be assigned to the true signature to be classified as 'successful', this result requires further consideration. Do the results change considerably, if the authors restrict the number of mutations to, say 50?

Reviewers' comments:

Notes:

Responses are in blue.

Red underlined text represents added material

Reviewer #1 (Remarks to the Author):

Temko et al present an in silico investigation exploring the putative relationship between mutational processes and driver mutation acquisition across multiple cancer types. In principle, I think this is quite an interesting analysis.

I would however, suggest some improvements:

- in the writing of the manuscript
- in description of the methods in order to ensure that all is absolutely transparent
- tidying up the signature extractions and assignments

We are grateful for the reviewer's thorough assessment of our manuscript, and glad that they find the work interesting.

Detailed critique:

INTRODUCTION

First para ok.

Second para ok.

Third para – this is an important para:

– improve description in this first sentence. It is apparent after reading the whole manuscript what you mean in this sentence but to a naïve reader, this sentence is entirely nonsensical. Keep the choice of words simple here and take the time to explain what you mean.

- Some punctuation missing,

o Traditionally (comma)

o Here (comma)

- Some word choices could be improved:

o Here we present not analyse

o We also explore not analyse

Response: We agree the description at the start of paragraph 3 could have been clearer, and have revised it to the below. The other small changes have been made.

The strength of selection experienced by a mutation is also expected to influence the frequency at which the mutation is detected in the patient population. If two mutations are equally likely to occur, we reason that the more strongly selected mutation will be found more frequently across cancers. Traditionally, it has been convenient to classify mutations found in cancer as drivers or passengers¹⁴, but it is likely that the effects of driver mutations actually lie on a continuum...

RESULTS

1.1 Testing for mutational signature and driver associations

• Para1: mutational signatures do not act on anything. Mutational processes are active. The signatures are the imprints left behind by those processes. I would change “acts” to “is active” or revamp the sentence completely.

Response: Thank you for pointing this out. We have revised the text as suggested.

We investigated the correlations between mutational signatures and recurrent driver mutations across cancer types. We reasoned that when a mutational process acts, it makes specific driver mutations, caused by a mutation in a specific channel enriched in the mutational signature of the process, more likely. We therefore tested for a difference in the levels of relative mutational process activity between cancers with and without specific driver mutations (Figure 1a). The use of signature and individual channel activity information was designed to increase the sensitivity and specificity of the approach. Where the activity of a mutational process was significantly higher in cancers with a mutation of interest compared to those without, we considered it supporting evidence for a causative relationship between the mutational process activity and the acquisition of the driver mutation.

1.2 • This sentence: “Where the activity of a mutational signature was significantly higher in cancers with a mutation of interest compared to those without, we considered it prima facie evidence of a causative relationship between the mutational signature activity and the presence of the driver mutation”.

This is the crux of this manuscript and is quite an exciting way to think but must be toned down. The reason is that it is never causative and only ever associative. Furthermore, there are signatures that are associated with each other, so you may be seeing a relationship that is secondary... so.. I would be careful about how this is worded.

Response: We thank the reviewer for this helpful comment. In our view, the associations between mutational signatures and driver mutations are consistent with *potentially* causal relationships (since we report cases where the specific channel that matches the specific driver mutation is enriched in the associated signature). Nevertheless we agree that this evidence alone is only associative, and so we cannot rule out other possible explanations of the data. We have revised the text throughout to make the scope of the claim clearer and to tone-down any claims of causality.

1.3 • The mutational signature analysis: How did you perform the extraction? Did you use SomaticSignatures R package using the 30 COSMIC signatures as an a priori, across the whole cohort of all those different tumor-types. (i.e. a global analysis) or did you do it tumor-type by tumor-type (local analysis). Doing a global analysis is a little dangerous for the next step - which is the assignments. It is well-known that the global analyses result in spilling of certain signatures into tissue-types that do not actually carry that mutational process biologically. This is a mathematical anomaly and various post-hoc (if slightly mad) filters have then been included to assign mutational signatures to individual samples in the past when global analyses have been performed. How did you deal with this? So

for example smoking-associated Signature 4 is well-known to spill into all sorts of tumor-types (e.g. prostate) just because a particular C>A dominant signature is present in that tumor-type (even though there is no smoking-associated process acting there. How did you exclude this problem? This is quite relevant to your analysis because you attribute a driver to a particular signature and imply causation. You need to make clear how you did your extraction and assignment.

Response: We apologise for the lack of clarity here. We did a version of the 'local analysis' the reviewer suggests. For each cancer type, the starting point for our analysis was the previously reported presence/absence activity of each of the 30 mutational signatures identified at the time of the analysis (as reported at: <http://cancer.sanger.ac.uk/cosmic/signatures>), and we used our regression method to assign an activity score to each of the signatures known to be active in the cancer type.

Action taken: We have updated the wording to clarify what was done in the Methods on page [10]:

The 96-channel context of each SNV was imputed using the R package 'SomaticSignatures'²⁷, and the total number of SNVs in each of the 96 channels was calculated for each sample. Non-synonymous mutations in driver genes were excluded. Mutational signatures were obtained from the Wellcome Trust Sanger Institute (<http://cancer.sanger.ac.uk/cosmic/signatures>) in April 2016. Information on the presence/absence of these signatures in individual cancer types was obtained from the same source. For whole exome data, signatures were re-scaled to the trinucleotide frequencies of the exome. Non-negative least squares regression, implemented in the R package 'nnls'²⁸, was used to assign a process activity for each signature reported as being present in the applicable cancer type. Signature activities were normalised for each sample to calculate the signature exposures. We note that analyses based on absolute signature activity gave broadly consistent results.

1.4 • It is also well-known that some of the mutational signatures such as Sig 5 is notoriously close (in terms of profile/cluster centroid) to other signatures (Signatures 3 and 8), and also not necessarily easily distinguished from Signature 1. Indeed, in some tissue types, it remains controversial whether Signature 5 is actually separate from Signature 1 unless performed in a particular mathematical way (a priori forcing to find these as separate signatures). Indeed, you show in Fig S1 that the classification accuracy is not good for Signature 5 when compared to other signatures. My question is then this: How can you be sure that your signature assignments are accurate for each driver mutation? Can you demonstrate to the reader that you have given this some consideration, even if one cannot be absolutely sure about the assignments? For signatures 2 and 13, I am certain that your assignments are pretty good, but I am less certain about Sig 5.

Response: We too were concerned about the accuracy of signature assignment. As is expected, assignment accuracy increases with the number of informative mutations (Figure S1). We chose a cut-off of 20 mutations per sample to allow us to include information from as many samples as possible, while still maintaining

reasonable accuracy of signature assignment. In this way, we aimed to maximise the sensitivity of our approach while maintaining specificity.

All three reviewers have asked about the possible effects of inaccurate signature assignments. To address this, we have repeated the analysis using a cut-off of 50 mutations. 37 of the original 43 associations were recovered with this higher cut-off, and 4 new associations were found. The high concordance of the results gives us additional confidence in the original results, and suggests that the loss of specificity using 20 mutations versus 50 mutations is limited.

However, as the reviewer highlights, we cannot completely rule out that some of the results are influenced by mis-assignment of mutational signatures. We have also added an extra caveat on the mutational signature assignments to reflect the similarities that some of the mutational signatures bear to each other. We have summarised these changes throughout the manuscript:

Page [4]:

Consequently, we excluded 1,153 samples with fewer than 20 mutations from this analysis, leaving 10,183 samples for further analysis. To test for potential signature mis-assignment, we also considered a more stringent cut-off of 50 mutations, which gave similar results (see Methods).

Page [10]:

We found that 20 mutations gave an average classification accuracy of 80% across signatures. As a result, we chose to use a cut-off of 20 mutations to strike a balance between including as many tumour samples as possible while still maintaining reasonable accuracy of signature assignment. We repeated the analysis of associations between driver mutations and mutational signatures using a cut-off of 50 mutations per sample for comparison. This analysis recovered 41 associations, of which 37 were also found using the 20 mutation cut-off

Page [8]:

Thirdly, some mutational signatures are similar to each other in composition, making it difficult to determine whether mutations are generated by one or more independent processes.

Mutational signatures shape driver mutation landscape

1.5 • Typo. Aflatoxin

Response: Corrected.

1.6 • Final para – this is sloppy use of terminology “linked to ageing”. Those signatures have been shown to be associated with age of diagnosis. That does not mean that they are pathognomonic of “ageing” (i.e. senescence). A correlation with age could be due to a number of factors – e.g. relative time or cell turnover – so to say that Signature 1 or 5 is linked to ageing is a step too far (though you are not the only ones to loosely use this description in this way).

Response: We agree that aging is an imprecise term, albeit one that is widely used, and have revised the wording on page [5]:

Six of the associations involved signatures that are known to correlate with age at diagnosis³

Detecting differential selection

1.7 When you are looking for evidence of differential selection, you do a comparison to a meta-sample which you have defined as an average for the relevant cluster-group. But for rare mutational processes e.g. MMR in breast cancer, a sample will end up in a cluster but effectively be an outlier in that cluster. So, when you determine the likelihood for seeing something at a particular sequence context, your expectation is based on an average but your observed refers to a specific mutation in a specific sample (which could be an outlier). You could over-call some of these as significant. Can your background model be improved to take into account per sample differences.

Response: This is an important point, and we are grateful to the reviewer for raising it (Reviewer 2, comment 2.5 raises a similar issue). In considering our response, we realised that it is possible to design the test for differential selection at the level of individual samples, rather than relying on cluster averages. Specifically, we have now used the Poisson binomial distribution to test the hypothesis that the mutation frequencies of two mutations are concordant with their likelihood of occurrence in each of the samples within a cancer type, based on a null hypothesis of equivalent selection. The sample-specific mutational probabilities are used as the input parameters in the Poisson binomial distribution. Reassuringly, we find that original results are robust to this improvement in the methodology. The new method is described in detail on page [11]:

Given two mutations M_1 and M_2 in a cancer type C , we identified the samples in which exactly one of the two mutations occurred. For each such sample S_i , we calculated p_{i1} and p_{i2} , the probability of the causal channels of M_1 and M_2 respectively occurring among the 96 mutation types, based on mutational signature exposures for the sample.

We used a Poisson binomial test to test whether the frequencies, m_1 and m_2 , of M_1 and M_2 were consistent with their relative probabilities of occurrence across the samples. Specifically, we modelled m_1 with the random variable X , where:

$X \sim \text{Poibin}(\mathbf{q}_1)$

$\mathbf{q}_1 = p_{i1} / (p_{i1} + p_{i2})$

We then used a two-tailed test to test whether the observed data differed from the predicted distribution.

Differential selection between pathogenic amino acid changes within a driver gene

1.8 • Second para needs to be improved in writing – this is quite an important para – and tie it up more precisely with your figures please. Fig 2B description comes after G13D but your figure is a comparison to G12D – it throws the reader. And I cannot see in Fig 2B how those show more strong selection than G13D at al.. Show the figure for G13D instead... seeing as you make the point of it in the manuscript. Or better still the KRAS G12C one... to highlight that all the symbols (same cluster) and mutation-type will be the same above the line – to emphasise that both mechanism and selection driving it. I do wonder whether there is a more intelligent way of demonstrating all this data – this is effectively a plot per comparison to each driver. The supplementary information is just hideous.

Response: In response to this and the following point, we have revised Figures 3-5 to directly show estimated differences in selection, in the form of modelled relative risks between mutations, including confidence intervals (and have similarly updated the supplementary material). The new figure ties in more closely with the text: KRAS G13D is labelled in panel A and the data for KRAS G12C in lung adenocarcinoma is shown, as suggested, in panel D.

1.9 Can you provide some sort of quantitative measure (with errors) of the effect of selection and present that instead (as a summarizing image/plot). That might give you more credence in demonstrating that you have a continuum of selective effects and help present the information more cohesively. Perhaps a heatmap where the 2D-distance from the grey dashed line is given for each mutation type in each gene. You will have 8 plots (one for each gene). Just a suggestion – the point is I think to think of a better way to present this information.

Response: Please see response to previous question. Revised figures and text detailed below.

Updated Figure 3 provided below:

Updates to methods on pages [10,11]:

Modelled relative risk

Given two mutations M_1 and M_2 , with probabilities of occurrence p_{i1} and p_{i2} in sample S_i , we model the probability q_{i1} , that M_1 is present given that either M_1 or M_2 is present by the following formula:

$$q_{i1} = (r_{1,2} * p_{i1}) / (r_{1,2} * p_{i1} + p_{i2})$$

where the parameter $r_{1,2}$ is the relative risk of mutation M_1 compared to mutation M_2 .

Defining I_1 and I_2 as the sets of sample numbers where M_1 and M_2 occurred, respectively, then the likelihood of the data, L , is given by:

$$L = \prod_{i \in I_1} q_i * \prod_{i \in I_2} (1 - q_i)$$

We used numerical methods to find the maximum likelihood estimate of $r_{1,2}$ for each pair of mutations in each tumour type, based on this formula. Bootstrapping with 100 iterations was used to find approximate confidence intervals around these estimates.

1.10 • Last para: I do not think that this is a fair comparison, across tissue-types because they will have very different prevalences of signatures.

Response: Our intention here was to highlight that the direction of apparent differential selection is consistent across cancer (tissue) types. Since our method controls for the levels of mutational process activity, in our view this provides

evidence for a (relatively) stable selective landscape across cancer types. We have added some wording to try to clarify the argument at this point on page [6].

Interestingly, the relative selective advantages of particular pathogenic mutations in each gene were broadly consistent across cancer types. Specifically, there were only 7/118 cases of differentially selected mutations where a mutation appeared selected more strongly than another in the same gene in one cancer type, but less strongly in another cancer type. Since our method controls for differences in mutational signature activity between cancer types, these results provide evidence to support the hypothesis that the mechanisms that underpin the selective advantage caused by a specific driver mutation are broadly uniform across tissue types.

Differential selection between mutationally exclusive driver genes

1.11 So I get this but it is not easy to follow this section and it is not easy to walk away with some form of over-arching sense of the message here without feeling like there has been only a token correction for mutational process. If you present data against a particular mutation (e.g. E545K), do you do it for the specific mutations in each cluster, or to all E545K mutations in samples that belong to any cluster for that tumor-type? I can't tell at the moment.

Response: We apologise for the lack of clarity here. As described above in response to point 1.7, we have now revised the methodology used in this analysis, to remove the need for clustering of samples. We have also simplified the narrative of this part of the manuscript.

Figure 1

1.12 a) Signature 5 H matrix to generate this must be contaminated with signature 1 very heavily. This does not look like the signature extraction and assignment has been very accurate.

b) I do not understand what this figure is trying to show... nor why it would be relevant. If I understand it correctly, it's simply showing the averaged 96-channel mutation profile across all signatures that are active for that tumor-type. It's very distracting and unhelpful.

c) Asterisk not explained. What is p-value. Please name statistic used to do the comparison (note: non-normal distributions).

Response: We have now significantly updated this figure in response to all three reviewers' comments. Of particular note to the points raised above, we have added the Q-values to the boxplots and named the statistical test in the figure legend. We apologise, the previous signature stated to be signature 5 was a different signature plotted in error. We provide the revised figure and legend below for ease of review.

Selected associations between driver mutations and mutational signature exposures within cancer types. Q-values shown are for Mann-Whitney U test. A) KRAS G12C and signature 4 in lung adenocarcinoma B) PIK3CA E545K and signature 2 in breast cancer C) FBXW7 R465C and Signature 15 in stomach cancer D) KRAS G12D and signature 26 in uterine carcinoma E) PTEN R130Q and signature 10 in uterine carcinoma F) APC R213X and signature 1 in colorectal cancer

1.13 Figure 2: OK

Figure 3 and 4 are ok but I think you could condense the information a little more, it's a lot of disparate plots and the message is slightly lost.

Response: As stated above, we have extensively reworked the analysis underlying figures 3 & 4, and used a new barchart format to present these new data.

SUPPLEMENTARY

1.14 • Fig S14 missing

1.15 • Description of Figure S15 and S16 is required. No idea what I am looking at.

Response: Apologies - these points have been corrected.

1.16 • Table S1, S3-S6 are OK

• Table S2 mutation frequency was calculated how? Need correction for frequency of triplets in genome/exome?

Response: The mutation frequency was calculated as the proportion of patients harbouring the mutation. We added this to give a sense of the proportion of patients that could be affected.

Action taken: We have changed the column title to 'tumour.frequency'

Reviewer #2 (Remarks to the Author):

Temko and colleagues present an analysis entitled "The effects of mutational processes and selection on driver mutations across cancer types". In this analysis the authors use publicly available whole exome and whole genome sequencing data from more than 10,000 tumour samples with respect to the relation of trinucleotide mutation spectra and recurrent coding base substitutions in known cancer driver genes.

The question to which extent the observed distributions of potential pathogenic hotspot mutations is caused by mutagenic processes alone, or due to positive selection, is important and has gathered considerable attention in the past years

The analysis presented consists of basically two parts. In the first part the authors assess the correlation of mutational signature activity and the presence of particular hotspots. This analysis is coupled with power calculations, which I highly commend.

We are grateful for the reviewer's accurate summary of our work, and glad that they find reason to commend our study.

2.1 There is however, a peculiarity in the sense that a positive correlation of a particular variant with the matched underlying mutational process generating this type of substitution rather suggests the absence of selection. Yet I guess what the authors are getting at is that the mutational processes are a strong determinant of the observed driver spectra. Wouldn't it be more informative, rather than testing for association to quantify the amount of explained variation, or an equivalent measure?

This is an interesting point. Under our simple model of mutation accumulation we expect the variation in frequency of a particular driver mutation within a cancer type to be influenced by mutational process activity, but also by other factors including the number of possible alternate driver mutations, and differential selection for those alternative mutations. Thus, we would argue that considering the variation alone doesn't necessary give more insight into the potentially causal relationship between mutational processes and driver mutation acquisition.

We see the first part of the analysis as playing a two-fold role, (i) providing individual examples of associations between mutational signatures and driver mutations that may be of interest for further research and (ii) providing support for the model used in the second part of the analysis, namely that mutational process activity, rather than selection, explains driver mutation frequency.

2.2 I also wondered how much of the association is explained by other effects such as transcriptional strand biases due to transcription couple repair, similarly replication strand biases for mismatch repair as well as age, gender and ethnicity. There could also possibly a higher order of the base context to be investigated.

Response: Our results demonstrate associations between mutational process activity (as read-out from the mutational signatures) and driver mutations. Nevertheless, we do accept the reviewer's assertion that the mutational processes are likely to co-vary with other important clinical and molecular features, and in some cases will be a proxy for these features. As a result, we do not claim that the relationships identified are necessarily mechanistic (and have suitably toned down any claims of causation in this revised manuscript).

That said, the fact that mutational process activity and driver mutations with significant associations are enriched for those cases with the strongest mechanistic basis for an association does give support for a mechanistic explanation in many of the cases. We have tried to make the exact nature of the claim clear in the text (see response to Reviewer 1 Comment 1.2) and have also added material describing the enrichment mentioned above. Page [4] and summarised below:

Our test for correlation between mutational processes and driver mutations focussed on processes which exhibit higher activity of the causal channel. This reduces the overall number of tests and increases the power to detect putative associations. However, to probe whether mutational processes and driver mutation acquisition are correlated in general, we repeated the analysis above without restricting the tests to signatures where the frequency of the causal channel was above average in the cancer type. An enrichment for positive associations between driver mutations and signatures where the underlying process has a higher than average activity of the causal channel would be indicative of a mechanistic relationship. Indeed, we found that 24 out of 37 significant associations had higher than average channel activity, compared to only 13 cases where the causal channel was lower than average ($P=5.5E-5$; Fisher's Exact Test; Figure S2; Table S3 for the full list of associations), supporting the notion that the respective mutational processes are responsible for the driver mutation. However, since our analysis is correlative, we cannot entirely rule out the possibility of other explanations for these associations. Despite this, the results above support a model whereby mutational processes play an important role in determining driver mutation spectrum.

2.3 Similarly, in my experience, the quantification of signature activity in individual samples can be very noisy with strong collinearity between inferred exposures. This uncertainty does not enter the test, except for the inclusion threshold of 20 mutations per sample, and I wondered whether this influences the outcomes. Related to this question is the choice of signatures - are the authors using any type of regularisation/model selection?

Response: We too were concerned about the accuracy of signature assignment. As explained in detail to review 1 comment 1.4, we have investigated the influence of increasing the number of mutations required for use in the signature assignment process from 20 mutations to 50 mutations and found that our assignments were acceptably robust (please see response to Reviewer 1 Comments 1.4 for details).

We interpret the comment about regularisation/model selection to refer to the issue of choosing which signatures were to consider for assignment in samples of

each cancer type. Here we have circumvented this issue by using only the signatures previously reported to be active in a cancer type (as stated at <http://cancer.sanger.ac.uk/cosmic/signatures>). Please see response to Reviewer 1 comment 1.3 for further details.

2.4 To demonstrate the validity of their test strategy it would be important to demonstrate the accuracy (QQ-plots) of their approach using simulations. Such simulations should recognise the aforementioned strand biases, transcription strength, replication timing, as well as the epigenetic associations with mutation rate that have been described by some of the authors before to make the background as realistic as possible. On genomes, this can be done using permutations within windows, for exomes one would need to establish a more elaborate way to keep the marginals fixed.

Response: This salient comment serves to highlight the difference between mutational signatures and an underlying mutational mechanism. As the reviewer's comment implies, mutational signatures are constructed from the average of mutation channel activity across the genome, and as such, local variation in mutational mechanism activity is not considered explicitly. Nevertheless, our analysis demonstrates an association between the (averaged) mutational signature and the presence/absence of a particular driver mutation. Since our data is associative, we cannot rule out the possibility that other factors – such as the molecular mechanisms listed by the reviewer - co-vary with the mutational processes and actually mechanistically cause some of the associations that we detect. In line with this, we have toned-down claims of causality throughout the manuscript.

In the second part of the analysis the authors investigate the frequencies of different hotspots in the same driver gene and in mutually exclusive pairs of driver genes. The authors cluster samples based on signature activity and then test for differences in hotspot frequencies using a binomial test as a measure of selection.

2.5 While this analysis is interesting it appears to be a bit of a black box and it would again be important to demonstrate the accuracy of this approach using realistic simulations (see above). I would be worried that the clustering missed important information that could explain some of the observed variance and therefore leads to false positives.

While dealing with an interesting scientific questions, in the absence of a systematic evaluation of the tests performed in this analysis I find it difficult to judge the reliability of the results.

Response: The mathematical basis of the analysis in the second part of the manuscript is to consider two 'equivalent' competing mutations M_1 and M_2 (e.g. two [typically] mutually exclusive mutations, that are in a general sense equally but redundantly 'necessary' for tumorigenesis – for example mutations of different residues of the same oncogene) that occur at rate u_1 and u_2

respectively. We calculate the 'relative risk' r of seeing M_1 instead of M_2 , given that one or other of the two mutations has occurred in the samples considered:

$$P(M_1|M_1 \cup M_2) = \frac{ru_1}{ru_1+u_2}$$

Previously we clustered samples to estimate cluster-specific mutation rates, but in response to the reviewers' comments about clustering missing important variation, we have now reworked the method to estimate a per-sample mutation rate. Reviewer 1 also critiqued the clustering approach, and we provide full details of the changes made in response to her/his comment 1.7. The results were robust to the change in methodology.

Reviewer #3 (Remarks to the Author):

Temko and colleagues used public cancer sequencing data to infer the independent effects of mutation and selection driver mutation complement. Their results suggest that selection is the key determinant of which driver alterations dominate in different cancer types. While an interesting topic, the manuscript lacks a coherent structure and this reviewer found it a little unclear, making it difficult to draw meaningful conclusions and provide a comprehensive review.

We are grateful for the reviewer's thorough critique below. We have reworked the flow of the manuscript, and hope that the reviewer finds the revised version easier to read.

A number of important points must be addressed before this article can be considered for publication:

3.1. The authors use both whole genome and predominantly whole exome sequencing analysis. Given that these samples will contribute vastly different numbers of mutations, the authors must demonstrate that this does not affect their results. For instance, if they simply use exome sequencing data, are the results concordant?

Response: The potential confounding of genomes versus exomes is an important point and we are grateful to the reviewer for raising it. To assess the effect of including whole genome samples on our results, we tested the effect of replacing the data from the 1,441 genomes with the data only from the exonic regions of those samples. We recovered 41/43 associations between driver mutations and mutational signatures in this way, and no new associations were found. In total 5/43 associations in the main analysis are in cancer types with at least 10 WGS samples, and in fact all 5 of these associations are all in liver cancer, for which 27% (305/1110) of samples are WGS. 3/5 of these associations are recovered and one still has a tendency towards significance ($P=0.071$). These results suggest that using WGS data in addition to WXS has a limited effect on our analysis.

We have summarised this analysis on page [3] and explained it in full in the methods page [11]:

Analysis using only exonic mutations for whole genome samples revealed similar relationships between mutational processes and driver mutations (see Methods).

Methods:

Our study used a combination of whole genome sequencing (WGS) and whole exome sequencing (WXS) data. The 1,441 whole WGS samples were distributed predominantly across 8/22 cancer types. In total five associations between mutational signatures and driver mutations were identified across these eight cancer types. All five of these associations were in liver cancer, where 27% of samples (305/1110) were WGS samples.

To assess the effect of using both whole genome and whole exome samples on our analysis, we analysed the effect on our results of replacing the WGS data only the exonic subset of mutations. We recovered 41/43 associations between driver mutations and mutational signatures and found no new associations, suggesting that using WGS data in addition to WXS has a limited effect on the analysis.

3.2. A central part of the analysis assumes accurate classification of mutational signatures. However, the methods are not entirely clear in this regard. A number of publications exist to identify mutational signatures and assign known signatures to samples – e.g. Alexandrov 2014, Rosenthal, 2016, Kim, 2016. Rather than using a published method to assign mutations to signatures the authors use non-negative least squares regression, implemented in R. The authors must demonstrate that their method is at least as good as published methods. Ideally, the authors should also implement published methods and demonstrate that their results are robust.

Response: The issue of accurate signature assignment is important, and indeed has been raised by all three reviewers. To address this reviewer's specific question about the performance of our method compared to other methods, we implemented `deconstructSigs`⁴ to assign signature exposures for each tumour for comparison. We used the candidate signatures for each tumour type reported at (cancer.sanger.ac.uk/cosmic/signatures), consistent with the main analysis. We found that for all signatures, relative signature exposures assigned by `deconstructSigs` were significantly correlated with exposures assigned by NNLS (correlations coefficient range 0.79-1.28; $P < 5 \times 10^{-154}$ each comparison). The fits of mutational signatures were slightly better on average using NNLS than `deconstructSigs` in terms of the root-mean square error (RMSE) (mean RMSE: 0.01 for NNLS, 0.01 for `deconstructSigs`; difference: 0.00021; see plots below). For comparison, using a uniform distribution of mutations across the 96 channels for each patient would give an RMSE of 25.2.

RMSE

RMSE

3.3. Associating a single point mutation to a particular signature can be a little problematic. The authors should discuss the limitations of this approach, and the fact that certain signatures are far less specific than others. For instance, while it may be relatively easy to assign a particular driver mutation to an APOBEC (Signature 2 or 13) context, discriminating between a C>T mutation at a CpG site in Signature 5 from one in Signature 1 is rather trickier.

Response: We do agree that the similarity that some mutational signatures bear to each other presents a challenge. The specific consequence for our study is the inference of causality: as the reviewer states, if a particular mutation could, with high likelihood, be caused by any of two or more mutational processes, we cannot be certain which process actually led to the accrual of the mutation. In recognition of this (and in response to other review comments), we have toned-down any claims of causality throughout the manuscript.

3.4. This manuscript would benefit from an additional Figure, preferably at the start, outlining the key questions that are being addressed and the methods used to address these. Such a figure may provide a point of reference for the reader, and, importantly, make the paper considerably easier to read. Indeed, without such a figure, this reviewer struggled to fully grasp the key biological questions addressed in the manuscript.

Response: Thank you for this helpful suggestion – we agree entirely. Our suggestion for a new Figure 1 explaining the outline of the analysis is provided below.

A) In the first part of the study, the effects of mutational signatures on driver mutation frequencies were investigated. For a driver mutation, the change was assigned to one of the 96 trinucleotide mutational channels (e.g. CTG>CCG). We hypothesised that mutational signatures in which that channel was higher than average would be over-represented in cancers with these mutations. We tested this hypothesis by comparing the levels of signatures in cancers harbouring the mutations to those in cancers that did not harbour the mutations.

B) In the second part of the study, we investigated the effects of mutational processes on the relative frequencies of specific pathogenic mutations in cancer driver genes. The causal channels of driver mutations with putative equivalent functional effects within cancer types were identified. We then tested whether observed frequencies of each driver mutation differed significantly from those expected based on mutational signatures, thus indicating differential selection between mutations in the same gene. Using a simple mathematical model, we transformed normalised measurements of mutation frequency into estimates of relative risk between mutations. This analysis was then extended to comparisons between mutations in different driver genes with apparently equivalent functional effects in a cancer type.

3.5. The authors normalize the signature activity to one for each sample, to calculate signature exposures. Such an approach may result in certain signatures 'drowning' out other signatures. For instance, elevated APOBEC mutational signature may result in a, relatively speaking, lower Signature 1, related to aging. However, this does not mean that the mutational process associated with aging necessarily contributed less to tumor development than in another tumor that does not exhibit APOBEC mutations. Thus, the authors should consider utilizing the total number of mutations associated with a signature rather than proportion of each signature present.

Response: Thank you for this comment. Under our model the probability of a driver mutation occurring in a tumour sample depends on the relative likelihood of the mutation occurring compared to other mutations. Formally, the probability of the mutation, p_i , is given by

$$p_i = \sum_j (e_j * s_{ij}),$$

Here e_j measures the normalised exposure of signature j ; s_{ij} is the probability of mutation channel i under signature j . We chose to use normalised signature exposures for this reason.

However, there are alternate models of the data under which absolute mutational signature activity may be more appropriate to use. To address this point we re-ran our analysis pipeline using absolute rather than relative mutational signature rates. In this adapted analysis we find 53 associations between mutational signatures and driver mutations. 81% (35/43) of the associations found in the main analysis are also identified here. The considerable overlap between the associations found by the two methods suggests that the use of normalised signature exposures compared to absolute rates does not have a major effect on the results. We have noted this in the methods page [9] of the revised manuscript.

3.6. The authors suggest 20 mutations give 'an average classification accuracy of 80% across signatures'. However, given that only 50% of the regression weights were required to be assigned to the true signature to be classified as 'successful', this result requires further consideration. Do the results change considerably, if the authors restrict the number of mutations to, say 50?

Response: In response to Reviewer 1 Comment 1.4 we have specifically addressed the robustness of signature assignment to the number of mutations required in each sample.

References

- 1 Castro-Giner, F., Ratcliffe, P. & Tomlinson, I. The mini-driver model of polygenic cancer evolution. *Nature reviews. Cancer* **15**, 680-685, doi:10.1038/nrc3999 (2015).
- 2 Vogelstein, B. *et al.* Cancer genome landscapes. *Science (New York, N.Y.)* **339**, 1546-1558, doi:10.1126/science.1235122 (2013).
- 3 Alexandrov, L. B. *et al.* Clock-like mutational processes in human somatic cells. *Nature genetics* **47**, 1402-1407 (2015).
- 4 Rosenthal, R., McGranahan, N., Herrero, J., Taylor, B. S. & Swanton, C. DeconstructSigs: delineating mutational processes in single tumors distinguishes DNA repair deficiencies and patterns of carcinoma evolution. *Genome Biol* **17**, 31 (2016).

REVIEWERS' COMMENTS:

Reviewer #2 (Remarks to the Author):

Temko et al have presented a revised manuscript. The main changes are:

- * a more careful presentation and discussion of underlying causality vs association
- * a comparison to results with more stringent criteria for (n=50) mutations per tumour
- * a comparison with a related method, `deconstructSigs`, showing little difference to their original `nns` approach to fit signatures.
- * a Poisson-Binomial test for testing differential selection

These changes address most of my comments and overall I think the results are fairly robust and interesting. However, based on the data presented, I wouldn't rule out some artefacts here and there.

As noted in my previous comment I'm not 100% sure whether the variance and bias from generating mutations and signature decomposition with possibly wrong signatures, wrong combinations of signatures in a given sample is correctly accounted for throughout the analysis. Increasing the number of mutations to 50 may help somewhat, but doesn't necessarily resolve all issues, so I'm not sure that the FDR levels aren't exceeded leading to a few more false positives than expected.

For example, while most of the results shown in Figure 1 seem plausible as they refer to signatures and base-substitutions with very distinct spectra, I wondered whether to really trust the KRAS G12D associations in Uterine cancer (Fig 1D) - are these samples truly MMR deficient or could it be a different process generating the underlying ACC>ATC mutations?

I still have similar concerns about the analysis of differential selection and the possible impact of signature decomposition; so overall I think the results point in the right direction, but need to be interpreted with a grain of salt.

Reviewer #3 (Remarks to the Author):

The authors have substantially improved their manuscript - improving the clarity and structure of the manuscript.

The manuscript could still benefit from a number of further modifications:

- 1) The authors understandably focus on the association between mutations in cancer genes and mutational signatures. However, I think more could be made of cases where the driver mutation does not fit the dominant mutational signature. E.g., the case of BRAF melanoma.
- 2) The authors should acknowledge that the concept of using mutational signatures to identify driver mutations and cancer genes is not novel. Indeed, this forms a significant part of analyses such as `MutSigCV2` and also, more recently, was shown to be essential when quantifying selection through `dNdS` analysis (Martincorena, 2017).
- 3). As a caveat, the authors should highlight that only point mutation drivers were considered, and should discuss whether a similar framework could be applied in copy number and methylation space.
- 4) The authors essentially make the assumption that there is no differential inherent mutability across a given gene. And, also no difference between two related genes (e.g. IDH1 and IDH2) -

ideally, the authors should also confirm that their observations are not also influenced by replication timing and expression (as outlined, e.g. in MutSigCV2).

5) I like the addition of Figure 1. But, I still think it potentially could be improved - perhaps by illustrating a couple of additional real examples.

Reviewers' comments:

Reviewer #2 (Remarks to the Author):

Temko et al have presented a revised manuscript. The main changes are:

- A more careful presentation and discussion of underlying causality vs association
- A comparison to results with more stringent criteria for (n=50) mutations per tumour
- A comparison with a related method, `deconstructSigs`, showing little difference to their original `nnls` approach to fit signatures.
- A Poisson-Binomial test for testing differential selection

These changes address most of my comments and overall I think the results are fairly robust and interesting.

Response: We are glad that the reviewer considers their comments largely addressed, and that they find our study interesting and judge it robust.

However, based on the data presented, I wouldn't rule out some artefacts here and there.

As noted in my previous comment I'm not 100% sure whether the variance and bias from generating mutations and signature decomposition with possibly wrong signatures, wrong combinations of signatures in a given sample is correctly accounted for throughout the analysis. Increasing the number of mutations to 50 may help somewhat, but doesn't necessarily resolve all issues, so I'm not sure that the FDR levels aren't exceeded leading to a few more false positives than expected.

Responses: We do agree that there is the potential for inaccuracies in the signature assignments. Indeed, our power analysis (Supplementary Figs. 1-3) showed that for 50 mutations per sample, the classification accuracy for many signatures is about 90%. We had previously acknowledged this limitation in the Discussion section (third to last paragraph beginning: "There are caveats to this analysis."), and have now added a new sentence towards the end of this paragraph explaining that the combined consequence of these caveats may lead to both false positives and negatives in signature associations and differential selection measures.

For example, while most of the results shown in Figure 1 seem plausible as they refer to signatures and base-substitutions with very distinct spectra, I wondered whether to really trust the KRAS G12D associations in Uterine cancer (Fig 1D) - are these samples truly MMR deficient or could it be a different process generating the underlying ACC>ATC mutations?

Response: To address this point we have analysed the microsatellite instability (MSI) status of those samples with and without this *KRAS G12D* mutation in

uterine carcinoma. MSI annotation was available from the GDC data portal for 16 of the 18 mutant samples (11 MSI-H, 1 MSI-L, 4 MSS). Annotation was available for 229/285 wild-type samples (59 MSI-H, 18 MSI-L, 152 MSS). Thus, MSI-H status was significantly enriched among mutant samples ($P=6.6E-4$). This analysis strongly suggests that the association between MMR-linked signatures and this mutation in uterine carcinoma reflects a genuine association with mismatch repair defects and does not result from signature mis-assignment.

I still have similar concerns about the analysis of differential selection and the possible impact of signature decomposition; so overall I think the results point in the right direction, but need to be interpreted with a grain of salt.

Response: The reviewer is of course correct that there is inaccuracy in our methodology – but this is inevitable in all mutational process deconvolution. We have extended our previous discussion of study limitations (in the Discussion section).

Secondly, we have relied on the assignment of signatures to individual samples and we note that some samples have relatively few mutations, making this assignment less accurate. Relatedly, in some cancer types, there are other active signatures that were not considered in this study. Where other signatures are present, the regression method used here can only approximate the signature contributions. Thirdly, some mutational signatures are similar to each other in composition, making it difficult to determine whether mutations are generated by one or more independent processes.

Reviewer #3 (Remarks to the Author):

The authors have substantially improved their manuscript - improving the clarity and structure of the manuscript.

Response: We are grateful for this positive assessment of our revised manuscript.

The manuscript could still benefit from a number of further modifications:

1) The authors understandably focus on the association between mutations in cancer genes and mutational signatures. However, I think more could be made of cases where the driver mutation does not fit the dominant mutational signature. E.g., the case of BRAF melanoma.

Response: This is an interesting point, and we thank the reviewer for the suggestion. We have now added a new paragraph commenting on cases where recurrent driver mutations do not match the dominant the mutational signature.

We note that there are cases of a common driver mutation in a cancer which do not match the dominant signature in that cancer type. For instance *BRAF V600E* (GTG>GAG) is not explained by signature 7, linked to ultraviolet (UV) exposure. Neither is *BRAF V600E* (GTG>GAG) explained by the dominant MMR-linked signature 6 in colorectal cancer. Similarly, *PTEN R130G* (ACG>AGG) is very common (>86% of samples) in uterine carcinoma but is not explained by signature 6, which is also dominant in this cancer. These cases point towards a key role for selection in addition to mutation in determining driver mutation incidence. In the next section, we turn to explore the role of selection in greater detail.

2) The authors should acknowledge that the concept of using mutational signatures to identify driver mutations and cancer genes is not novel. Indeed, this forms a significant part of analyses such as MutSigCV2 and also, more recently, was shown to be essential when quantifying selection through dNdS analysis (Martincorena, 2017).

Response: We thank the reviewer for emphasising this, and have now added a paragraph to the discussion referencing these two methods, and explaining the difference our approach compared to these methods.

Previous methods have taken into account the variability of mutation rates between mutation types to infer selection¹⁻³. Through the use of the dN/dS measure (the ratio of non-synonymous mutations to synonymous mutations within a gene) two of these studies have gone some way to quantifying the effects of selection experience by individual genes, beyond a binary distinction between driver genes and passenger genes. However, all these studies quantify selection at the level of genes, rather than the level of individual mutations. To the best of our knowledge, this study is the first to take into account the variation

in background mutation rates to quantify selection experienced by individual nucleotide driver mutations in a pan-cancer analysis.

3). As a caveat, the authors should highlight that only point mutation drivers were considered, and should discuss whether a similar framework could be applied in copy number and methylation space.

Response: This is an interesting suggestion, that we have now made in the discussion:

In this study we have analysed the influence of mutation and selection on single nucleotide alteration frequencies. In theory a similar analysis could be carried out for copy number alterations or methylation changes. At present the limited knowledge of pan-cancer mutational signatures involving these types of change make this challenging. However, the expected publication of greater numbers of whole genome sequencing and methylation studies could make this possible in the near future.

4) The authors essentially make the assumption that there is no differential inherent mutability across a given gene. And, also no difference between two related genes (e.g. IDH1 and IDH2) - ideally, the authors should also confirm that their observations are not also influenced by replication timing and expression (as outlined, e.g. in MutSigCV2).

Response: The reviewer is correct that we assume constant mutation rates across a gene, and across the sets of 'equivalent genes' that we compare in the final part of our analysis. The main reason for this is one of pragmatism: the reviewer will appreciate that there is scant data available to evaluate the assumption with sufficient accuracy. For our multi-gene comparison, we have now conducted some new analysis to give a sense of the probable impact of this assumption (reported below)

We downloaded gene-level replication timing, expression level and non-coding mutation rate data from the MutSigCV manuscript¹, and assigned each of the 16,918 genes with complete information to a replication timing decile and an expression decile. Among the gene sets we analysed for differential selection, *CTNNB1* and *APC* and *KRAS*, *BRAF* and *NRAS* were very similar in terms of their replication timing and expression levels (figure below), equating to differences in the per-gene mutation rate of <10%. *IDH1* and *IDH2* differed more: the mutation rate for *IDH1* was rate 34% higher than the rate for genes similar to *IDH2*. However, we note that in glioma, among the *IDH1* and *IDH2* mutations that we analysed that occur at least once, differential selection between *IDH1* and *IDH2* mutations is ~4-fold on average. Together, we argue that this analysis highlights the potential inaccuracy in the quantification of the precise selective difference between driver mutations, but nevertheless illustrates that our inferences of large selective differences between equivalent driver mutations are most likely robust.

We have added a comment to the manuscript to reflect this analysis.

We note that based on data from MutSigCV¹, genes similar to *KRAS*, *BRAF* and *NRAS*, in replication timing and expression have similar background mutation rates, as do genes similar to *APC* and *CTNNB1*. The background mutation rates of genes similar to *IDH1* and *IDH2* show a greater difference, but one that is still smaller than selective differences inferred between these genes.

5) I like the addition of Figure 1. But, I still think it potentially could be improved - perhaps by illustrating a couple of additional real examples.

Response: We have further revised the figure to better explain the methodology in part 2 (differential selection) by adding in a panel showing how signature information was utilised.

References

- 1 Lawrence, M. S. *et al.* Mutational heterogeneity in cancer and the search for new cancer-associated genes. *Nature* **499**, 214-218 (2013).

- 2 Martincorena, I. *et al.* Universal Patterns of Selection in Cancer and Somatic Tissues. *Cell* **171**, 1029-1041 e1021 (2017).
- 3 Weghorn, D. & Sunyaev, S. Bayesian inference of negative and positive selection in human cancers. *Nature genetics* **49**, 1785-1788, doi:10.1038/ng.3987 (2017).